# A Pseudo-Metric between Probability Distributions based on Depth-Trimmed Regions

**Guillaume Staerman**                                                                    *guillaume.staerman@inria.fr*
*LTCI, Télécom Paris, Institut Polytechnique de Paris*

**Pavlo Mozharovskyi**                                                            *pavlo.mozharovskyi@telecom-paris.fr*
*LTCI, Télécom Paris, Institut Polytechnique de Paris*

**Pierre Colombo**                                                                *pierre.colombo@centralesupelec.fr*
*Equall.ai and MICS, CentraleSupélec, Université Paris-Saclay*

**Stéphan Clémençon**                                                            *stephan.clemencon@telecom-paris.fr*
*LTCI, Télécom Paris, Institut Polytechnique de Paris*

**Florence d'Alché-Buc**                                                          *florence.dalche@telecom-paris.fr*
*LTCI, Télécom Paris, Institut Polytechnique de Paris*

**Reviewed on OpenReview:** *https://openreview.net/forum?id=QySD5r7PeE*

## Abstract

The design of a metric between probability distributions is a longstanding problem motivated by numerous applications in machine learning. Focusing on probability distributions in the Euclidean space $\mathbb{R}^d$, we introduce a novel pseudo-metric between probability distributions by leveraging the extension of univariate quantiles to multivariate spaces. Data depth is a nonparametric statistical tool that measures the centrality of any element $x \in \mathbb{R}^d$ with respect to (w.r.t.) a probability distribution or a dataset. It is a natural median-oriented extension of the cumulative distribution function (cdf) to the multivariate case. Thus, its upper-level sets—the depth-trimmed regions—give rise to a definition of multivariate quantiles. The new pseudo-metric relies on the average of the Hausdorff distance between the depth-based quantile regions for each distribution. Its good behavior regarding major transformation groups, as well as its ability to factor out translations, are depicted. Robustness, an appealing feature of this pseudo-metric, is studied through the finite sample breakdown point. Moreover, we propose an efficient approximation method with linear time complexity w.r.t. the size of the dataset and its dimension. The quality of this approximation and the performance of the proposed approach are illustrated in numerical experiments.

## 1 Introduction

Metrics or pseudo-metrics between probability distributions have attracted a long-standing interest in information theory (Kullback, 1959; Rényi, 1961; Csiszàr, 1963; Stummer & Vajda, 2012), probability theory and statistics (Billingsley, 1999; Sriperumbudur et al., 2012; Panaretos & Zemel, 2019; Rachev, 1991). While they serve many purposes in machine learning (Cha & Srihari, 2002; MacKay, 2003), they are of crucial importance in automatic evaluation of natural language generation (see e.g. Kusner et al., 2015; Zhang et al., 2019), especially when leveraging deep contextualized embeddings such as the popular BERT (Devlin et al., 2018). Yet designing a measure to compare two probability distributions is a challenging research field. This is certainly due to the inherent difficulty in capturing in a single measure typical desired properties such as:

(i) metric or pseudo metric properties, (ii) invariance under specific geometric transformations, (iii) efficient computation, and (iv) robustness to contamination.

One can find in the literature a vast collection of discrepancies between probability distributions that rely on different principles. The $f$-divergences (Csiszàr, 1963) are defined as the weighted average by a well-chosen function $f$ of the odds ratio between the two distributions. They are widely used in statistical inference but are, by design, ill-defined when the supports of both distributions do not overlap, which is a significant limitation in many applications. Integral Probability Metrics (IPMs; Sriperumbudur et al., 2012) are based on a variational definition of the metric, i.e. the maximum difference in expectation for both distributions calculated over a class of measurable functions and give rise to various metrics (Maximum Mean Discrepancy (MMD), Dudley's metric, $L_1$-Wasserstein Distance) depending on the choice of this class. However, except in the case of MMD, which enjoys a closed-form solution, the variational definition raises issues in computation. From the side of Optimal transport (OT) (see Villani, 2003; Peyré & Cuturi, 2019), the $L_p$-Wasserstein distance is based on a ground metric able to take into account the geometry of the space on which the distributions are defined. Its ability to handle non-overlapping support and appealing theoretical properties make OT a powerful tool, mainly when applied to generative models (Arjovsky et al., 2017), domain adaptation (Courty et al., 2014; Courty et al., 2017), realign datasets in natural sciences (Janati et al., 2019; Schiebinger et al., 2019) or automatic text evaluation (Zhao et al., 2019; Colombo et al., 2021a).

In this work, we adopt another angle. Focusing on probability distributions in the Euclidean space $\mathbb{R}^d$, we propose to consider a new metric between probability distributions by leveraging the extension of univariate quantiles to multivariate spaces. The notion of quantile function is an interesting ground to build a comparison between two probability measures as illustrated by the closed-form of the Wasserstein distance defined over $\mathbb{R}$. However, given the lack of natural ordering on $\mathbb{R}^d$ as soon as $d > 1$, extending the concept of univariate quantiles to the multivariate case raises a real challenge. Many extensions have been proposed in the literature, such as minimum volume sets (Einmmahl & Mason, 1992), spatial quantiles (Koltchinskii & Dudley, 1996) or data depth (Tukey, 1975). The latter offers different ways of ordering multivariate data regarding a probability distribution. Precisely, *data depths* are non-parametric statistics that determine the centrality of any element $x \in \mathbb{R}^d$ w.r.t. a probability measure. They provide a multivariate ordering based on topological properties of the distribution, allowing it to be characterized by its location, scale or shape (see, e.g. Mosler, 2013 or Chapter 2 of Staerman, 2022 for a review). Several data depths were subsequently proposed, such as convex hull peeling depth (Barnett, 1976), simplicial depth (Liu, 1990), Oja depth (Oja, 1983) or zonoid depth (Koshevoy & Mosler, 1997) differing in their properties and applications. With a substantial body of literature devoted to its computation, recent advances allow for fast exact (Pokotylo et al., 2019) and approximate (Dyckerhoff et al., 2021) computation of several depth notions. The desirable properties of data depth, such as affine invariance, continuity w.r.t. its arguments, and robustness (Zuo & Serfling, 2000) make it an important tool in many fields. Today, in its variety of notions and applications, data depth constitutes a versatile methodology (Mosler & Mozharovskyi, 2021) that has been successfully employed in a variety of machine learning tasks such as regression (Rousseeuw & Hubert, 1999; Hallin et al., 2010), classification (Li et al., 2012; Lange et al., 2014), anomaly detection (Serfling, 2006; Rousseeuw & Hubert, 2018; Staerman et al., 2020) and clustering (Jörnsten, 2004).

This paper presents a new discrepancy measure between probability distributions, well-defined for non-overlapping supports, that leverages the interesting features of data depths. This measure is studied through the lens of the previously stated properties, yielding the contributions listed below.

**Contributions:**

- A new discrepancy measure between probability distributions involving the upper-level sets of data depth is introduced. We show that this measure defines a pseudo-metric in general. Its good behavior regarding major transformation groups, as well as its ability to factor out translations, are depicted. Its robustness is investigated through the concept of finite sample breakdown point.

- An efficient approximation of the depth-trimmed regions-based pseudo-metric is proposed for convex depth functions such as halfspace and projection. This approximation relies on a nice feature of the Hausdorff distance when computed between convex bodies.

- The behavior of this algorithm regarding its parameters is studied through numerical experiments, which also highlight the by-design robustness of the depth-trimmed regions based pseudo-metric. Applications to robust clustering of images and automatic evaluation of natural language generation (NLG) show the benefits of this approach when benchmarked with state-of-the-art probability metrics.

## 2 Background on Data Depth

In this section, we recall the concept of statistical *data depth* function and its attractive theoretical properties for clarity. Here and throughout, the space of all probability measures on $\mathbb{R}^d$ with $d \in \mathbb{N}^*$ is denoted by $\mathcal{M}_1(\mathbb{R}^d)$. By $g_\sharp$ we denote the push-forward operator of the function $g$. Introduced by Tukey (1975), the concept of data depth initially extends the notion of median to the multivariate setting. In other words, it measures the centrality of any element $x \in \mathbb{R}^d$ regarding a probability distribution (respectively, a dataset). Formally, a data depth is defined as follows:

$$
\begin{aligned}
D: \quad \mathbb{R}^d \times \mathcal{M}_1(\mathbb{R}^d) &\longrightarrow [0,1], \\
(x, \rho) &\longmapsto D(x, \rho).
\end{aligned}
\tag{1}
$$

We denote by $D(x, \rho)$ (or $D_\rho(x)$ for brevity) the depth of $x \in \mathbb{R}^d$ w.r.t. $\rho \in \mathcal{M}_1(\mathbb{R}^d)$. The higher $D(x, \rho)$, the deeper it is in $\rho$. The depth-induced median of $\rho$ is then defined by the set attaining $\sup_{x \in \mathbb{R}^d} D(x, \rho)$. Since data depth naturally and in a nonparametric way defines a pre-order on $\mathbb{R}^d$ w.r.t. a probability distribution, it can be seen as a centrality-based alternative to the cumulative distribution function (cdf) for multivariate data. For any $\alpha \in [0, 1]$, the associated $\alpha$-depth region of a depth function is defined as its upper-level set:

$$
D_\rho^\alpha = \left\{ x \in \mathbb{R}^d, \, D_\rho(x) \geq \alpha \right\}.
$$

It follows that depth regions are nested, i.e. $D_\rho^{\alpha'} \subseteq D_\rho^\alpha$ for any $\alpha < \alpha'$. These depth regions generalize the notion of quantiles to a multivariate distribution.

A depth function's relevance to capturing information about a distribution relies on the statistical properties it satisfies. Such properties have been thoroughly investigated in Liu (1990); Zuo & Serfling (2000) and Dyckerhoff (2004) with slightly different sets of axioms (or postulates) to be satisfied by a proper depth function. In this paper, we restrict to *convex depth functions* (Dyckerhoff, 2004) mainly motivated by recent algorithmic developments including theoretical results (Nagy et al., 2020) as well as implementation guidelines (Dyckerhoff et al., 2021).

The general formulation (1) opens the door to various possible definitions. While these differ in theoretical and practically related properties such as robustness or computational complexity (see Mosler & Mozharovskyi, 2021 for a detailed discussion), several postulates have been developed throughout the recent decades the "good" depth function should satisfy. Formally, a function $D$ is called a *convex depth function* if it satisfies the following postulates:

**D1** (AFFINE INVARIANCE) $D(g(x), g_\sharp \rho) = D(x, \rho)$ holds for $g : x \in \mathbb{R}^d \mapsto Ax + b$ with any non-singular matrix $A \in \mathbb{R}^{d \times d}$ and any vector $b \in \mathbb{R}^d$.

**D2** (VANISHING AT INFINITY) $\lim_{||x|| \to \infty} D_\rho(x) = 0$.

**D3** (UPPER SEMICONTINUITY) $\left\{ x \in \mathbb{R}^d \, D_\rho(x) < \alpha \right\}$ is an open set for every $\alpha \in (0, 1]$.

**D4** (QUASICONCAVITY) For every $\lambda \in [0, 1]$ and $x, y \in \mathbb{R}^d$, $D_\rho(\lambda x + (1 - \lambda)y) \geq \min\{D_\rho(x), D_\rho(y)\}$.

While (**D1**) is useful in applications providing independence w.r.t. measurement units and coordinate system, (**D2**) and (**D3**) appear as natural properties since data depth is a (center-outward) generalization of cdf. Limit values vanish due to median-oriented construction. (**D4**) allows to preserve the original center-outward ordering goal of data depth and induces convexity of the depth regions. Furthermore, it is easy to see that

(**D1**–**D4**) respectively yield properties of affine equivariance, boundedness, closedness and convexity of the central regions $D_\rho^\alpha$ (Dyckerhoff, 2004). Thanks to (**D2**–**D4**), if $\alpha > 0$, non-empty regions associated to convex depth functions are convex bodies (compact convex set in $\mathbb{R}^d$).

Below we recall two convex depth functions satisfying (**D1**–**D4**) that will be used throughout the paper: the halfspace depth (Tukey, 1975) and the projection depth (Liu, 1992), which are probably the most studied in the literature. For this, let $\mathbb{S}^{d-1}$ be the unit sphere in $\mathbb{R}^d$ and $X$ a random variable defined on a certain probability space $(\Omega, \mathcal{A}, \mathbb{P})$ that takes values in $\mathcal{X} \subset \mathbb{R}^d$ following distribution $\rho$. The halfspace depth of a given $x \in \mathbb{R}^d$ w.r.t. $\rho$ is defined as the smallest probability mass that can be contained in a closed halfspace containing $x$:

$$HD_\rho(x) = \inf_{u \in \mathbb{S}^{d-1}} \mathbb{P}\left(\langle u, X \rangle \leq \langle u, x \rangle\right).$$

Projection depth, being a monotone transform of the Stahel-Donoho outlyingness (Donoho & Gasko, 1992; Stahel, 1981), is defined as follows:

$$PD_\rho(x) = \left(1 + \sup_{u \in \mathbb{S}^{d-1}} \frac{|\langle u, x \rangle - \mathrm{med}(\langle u, X \rangle)|}{\mathrm{MAD}(\langle u, X \rangle)}\right)^{-1},$$

where med and MAD stand for the univariate median and median absolute deviation from the median, respectively.

**Remark 2.1.** *Data depth functions have connections with the density function in particular cases. Indeed, for elliptical distributions, the level sets of any data depth satisfying (**D1**–**D4**) are concentric ellipsoids with the same center, and orientation as the density level sets (Liu & Singh, 1993). The density is a local measure assigning the score of an element as the probability mass in an infinitesimal neighborhood. In contrast, data depths are global measures of ordering taking into account the whole distribution to assign a score to an element and are thus not equivalent to the density for general distributions. However, they provide interesting alternatives in many applications, such as anomaly detection (see e.g. Staerman et al., 2021b). For example, the density will assign a zero score to every $x \in \mathbb{R}^d$ far from a concentrated group of observations regardless of the distance. At the same time, the projection depth described above will be able to rank these "outliers" depending on how it moves away from them.*

## 3   A Pseudo-Metric based on Depth-Trimmed Regions

In this section, we introduce the depth-based pseudo-metric and study its properties. We consider depth regions possessing the same probability mass to compare those from different probability distributions fairly. Following Paindaveine & Bever (2013), we denote by $\alpha : (\beta, \rho) \in [0, 1] \times \mathcal{M}_1(\mathbb{R}^d) \longmapsto \alpha(\beta, \rho) \in [0, 1]$ the highest level such that the probability mass of the depth-trimmed region at this level is at least $\beta$. Precisely, for any pair $(\beta, \rho) \in [0, 1) \times \mathcal{M}_1(\mathbb{R}^d)$:

$$\alpha(\beta, \rho) = \sup\{\gamma \in [0, 1] : \ \rho\left(D_\rho^\gamma\right) > \beta\}. \tag{2}$$

In the remainder of this paper, when the quantity $\alpha(\beta, \rho)$ will be associated with depth regions of $\rho$, the second argument of the function $\alpha(\cdot, \cdot)$ will be omitted, for notation simplicity. It is worth mentioning that $D_\rho^{\alpha(\beta')} \subseteq D_\rho^{\alpha(\beta)}$ for any $\beta > \beta'$, since $\beta \mapsto \alpha(\beta, \rho)$ is a monotone decreasing function. Thus, $D_\rho^{\alpha(\beta)}$ is the smallest depth region with probability larger than or equal to $\beta$ and can be defined in an identical way as:

$$D_\rho^{\alpha(\beta)} = \bigcap_{\gamma \in \Gamma_\rho(\beta)} D_\rho^\gamma,$$

where $\Gamma_\rho(\beta) = \{\zeta \in [0, 1] : \ \rho\left(D_\rho^\zeta\right) > \beta\}$. The strict inequalities in (2) and in the definition of $\Gamma_\rho(\beta)$ eliminate cases where the supremum does not exist. Indeed, when $\beta = 0$, the depth region is then an infinitesimal set with a probability higher than zero. To the best of our knowledge, the supremum exists (without necessarily

being unique) in the case of the halfspace depth (Rousseeuw & Rutz, 1999) and the projection depth (Zuo, 2003) under mild assumptions. The set $\{D_\rho^{\alpha(\beta)}, \ \beta \in [0, 1 - \varepsilon], \ \varepsilon \in (0, 1]\}$ where each region probability mass is equal to $\beta$ then defines quantile regions of $\rho$.

Let $\mu, \nu$ be two probability measures on $\mathcal{X}, \mathcal{Y} \subset \mathbb{R}^d$ respectively. Denote by $d_{\mathcal{H}}(A, B)$ the Hausdorff distance between the sets $A$ and $B$. The pseudo-metric between probability distributions $\mu$ and $\nu$ based on the depth-trimmed regions is defined as follows.

**Definition 3.1.** *Let $\varepsilon \in (0, 1]$ and $p \in (0, \infty)$, for all pairs $(\mu, \nu)$ in $\mathcal{M}_1(\mathcal{X}) \times \mathcal{M}_1(\mathcal{Y})$, the depth-trimmed regions $(DR_{p,\varepsilon})$ discrepancy measure between $\mu$ and $\nu$ is defined as*

$$DR_{p,\varepsilon}^p(\mu, \nu) = \int_0^{1-\varepsilon} d_{\mathcal{H}} \left( D_\mu^{\alpha(\beta)}, D_\nu^{\alpha(\beta)} \right)^p \ \mathrm{d}\beta. \tag{3}$$

Our discrepancy measure relies on the Hausdorff distance averaged over depth-trimmed regions with the same probability mass w.r.t. each distribution. Properties (**D2–D3**) ensure that for every $0 \leq \beta < 1$, $D_\mu^{\alpha(\beta)}$ is a non-empty compact subset of $\mathbb{R}^d$ leading to a well-defined discrepancy measure. Observe that the parameter $\varepsilon$ can be considered as a robustness tuning parameter. Indeed, choosing higher $\varepsilon$ amounts to ignore the larger upper-level sets of data depth function, i.e. the tails of the distributions, see Sections 3.2 and 5.1.

**Remark 3.2.** *Data depths provide robustness to (3) together with the $\varepsilon$-trimming. Indeed, data depths such as the three previously introduced in Section 2 exhibit attractive robustness properties. The asymptotic breakdown point of the halfspace median is higher than $1/(d+1)$. In contrast, the projection median is known to have a breakdown point equal to $1/2$ (Donoho & Gasko, 1992; Ramsay et al., 2019).*

**Remark 3.3.** *When $d = 1$, the $L_p$-Wasserstein distance enjoys an explicit expression involving quantile and distribution functions. Let $X^1 \sim \mu_1$, $Y^1 \sim \nu_1$ be two random variables where $\mu_1, \nu_1$ are univariate probability distributions. Denoting by $F_{X^1}^{-1}$ the quantile function of $X^1$, the $L_p$-Wasserstein distance can be written as*

$$W_p^p(\mu_1, \nu_1) = \int_0^1 |F_{X^1}^{-1}(q) - F_{Y^1}^{-1}(q)|^p \ \mathrm{d}q. \tag{4}$$

*Since data depth and its central regions are extensions of cdf and quantiles to dimension $d > 1$, $DR_{p,\varepsilon}$ is then a possible (center-outward) generalization of (4) to higher dimensions. When $DR_{p,\varepsilon}$ is associated with the halfspace depth, a simple calculus (see Lemma A.3 in the Appendix for mathematical details) leads to*

$$DR_{p,\varepsilon}^p(\mu_1, \nu_1) = 2 \int_{\varepsilon/2}^{1/2} \max \left\{ |F_{X^1}^{-1}(q) - F_{Y^1}^{-1}(q)|^p, \ |F_{X^1}^{-1}(1-q) - F_{Y^1}^{-1}(1-q)|^p \right\} \ \mathrm{d}q.$$

*Thus, $W_p^p(\mu_1, \nu_1) \leq \lim_{\varepsilon \to 0} DR_{p,\varepsilon}^p(\mu_1, \nu_1)$ in general where the equality holds for symmetric distributions.*

### 3.1 Metric Properties

We now investigate to which extent the proposed discrepancy measure satisfies the metric axioms. As a first go, we show that $DR_{p,\varepsilon}$ fulfills most conditions. However, it does not define distance in general.

**Proposition 3.4** (METRIC PROPERTIES)**.** *For any convex data depth, $DR_{p,\varepsilon}$ is positive, symmetric and satisfies triangular inequality but the entailment $DR_{p,\varepsilon}(\mu, \nu) = 0 \implies \mu = \nu$ does not hold in general.*

Thus, $DR_{p,\varepsilon}$ defines a pseudo-metric rather than a distance. Based on distance, the proposed discrepancy measure preserves isometry invariance, as stated in the following proposition.

**Proposition 3.5** (ISOMETRY INVARIANCE)**.** *Let $A \in \mathbb{R}^{d \times d}$ be a non-singular matrix and $b \in \mathbb{R}^d$. Define the isometry mapping $g : x \in \mathbb{R}^d \mapsto Ax + b$ with $AA^\top = I_d$, then it holds:*

$$DR_{p,\varepsilon}(g_\sharp \mu, g_\sharp \nu) = DR_{p,\varepsilon}(\mu, \nu),$$

*where $g_\sharp \mu$ is the push-forward of $\mu$ by $g$. In particular, it ensures invariance of $DR_{p,\varepsilon}$ under translations and rotations.*

Although formulas (3) and (4) are based on the same spirit, there are no apparent reasons why the proposed pseudo-metric should have the same behavior as the Wasserstein distance. It is the purpose of Proposition 3.6 to investigate the ability to factor out translations, for $DR_{2,\varepsilon}$ associated with the halfspace depth, giving a positive answer for the case of two Gaussian distributions with equal covariance matrices.

**Proposition 3.6** (TRANSLATION CHARACTERIZATION). *Consider $X, Y$ two random variables following $\mu \in \mathcal{M}_1(\mathcal{X})$ and $\nu \in \mathcal{M}_1(\mathcal{Y})$ with expectations $\mathbf{m}_1, \mathbf{m}_2$ and variance-covariance matrices $\mathbf{\Sigma}_1, \mathbf{\Sigma}_2$ respectively. Denoting by $\mu^*, \nu^*$ the centered versions of $\mu, \nu$, it holds:*

$$\left| DR_{2,\varepsilon}^2(\mu, \nu) - DR_{2,\varepsilon}^2(\mu^*, \nu^*) - ||\mathbf{m}_1 - \mathbf{m}_2||^2 \right| \leq 2\, DR_{1,\varepsilon}(\mu^*, \nu^*) ||\mathbf{m}_1 - \mathbf{m}_2||.$$

*Now, let $\mu \sim \mathcal{N}(\mathbf{m}_1, \mathbf{\Sigma}_1)$ and $\nu \sim \mathcal{N}(\mathbf{m}_2, \mathbf{\Sigma}_2)$. Then it holds:*

$$\left| DR_{1,\varepsilon}(\mu, \nu) - ||\mathbf{m}_1 - \mathbf{m}_2|| \right| \leq C_\varepsilon \sup_{u \in \mathbb{S}^{d-1}} \left| \sqrt{u^\top \mathbf{\Sigma}_1 u} - \sqrt{u^\top \mathbf{\Sigma}_2 u} \right|,$$

*where $C_\varepsilon = \int_0^{1-\varepsilon} \left| \Phi^{-1}(1 - \alpha(\beta)) \right| \, d\beta$ with $\Phi$ the cdf of the univariate standard Gaussian distribution.*

Following Proposition 3.6: when $\mathbf{\Sigma}_1 = \mathbf{\Sigma}_2$, one has $DR_{2,\varepsilon}(\mu, \nu) = DR_{1,\varepsilon}(\mu, \nu) = ||\mathbf{m}_1 - \mathbf{m}_2||$ for any $\mu \sim \mathcal{N}(\mathbf{m}_1, \mathbf{\Sigma}_1)$ and $\nu \sim \mathcal{N}(\mathbf{m}_2, \mathbf{\Sigma}_2)$ providing a closed-form expression in the Gaussian case. This proposition shows that $DR_{2,\varepsilon}$ can factor out translations in a similar way as Wasserstein distance if $DR_{1,\varepsilon}(\mu^*, \nu^*)$ is zero. Furthermore, it is clear that if $DR_{1,\varepsilon}(\mu^*, \nu^*) = 0$ then $DR_{2,\varepsilon}(\mu^*, \nu^*)$ is zero too.

## 3.2 Robustness

In this part, we explore the robustness of the proposed distance, associated with the halfspace depth, given the finite sample breakdown point (BP; Donoho, 1982; Donoho & Hubert, 1983). This notion investigates the smallest contamination fraction under which the estimation breaks down in the worst case. Considering a sample $\mathcal{S}_n = \{X_1, \dots, X_n\}$ composed of i.i.d. observations drawn from a distribution $\mu$ with empirical measure $\hat{\mu}_n = (1/n) \sum_{i=1}^n \delta_{X_i}$, the finite sample breakdown point of $DR_{p,\varepsilon}$ w.r.t. $\mathcal{S}_n$, denoted by $BP(DR_{p,\varepsilon}, \mathcal{S}_n)$ is defined as:

$$\min \left\{ \frac{o}{n+o} : \sup_{Z_1, \dots, Z_o} DR_{p,\varepsilon}(\hat{\mu}_{n+o}, \hat{\mu}_n) = +\infty \; ; \; o \in \mathbb{N}^* \right\},$$

where $\hat{\mu}_{n+o} = \frac{1}{n+o} \left( \sum_{i=1}^n \delta_{X_i} + \sum_{j=1}^o \delta_{Z_j} \right)$ is the "concatenate" empirical measure between $X_1, \dots, X_n$ and the contamination sample $Z_1, \dots, Z_o$ with $o \in \mathbb{N}^*$. It is well known that the extremal regions of the halfspace depth are not robust while its central regions are rather stable under contamination (Donoho & Gasko, 1992). Fortunately, by construction, the parameter $\varepsilon$ allows us to ignore these extremal depth regions and thus ensure the robustness of the depth-trimmed regions distance. Based on the results of Donoho & Gasko (1992) and Nagy & Dvořák (2021), the following proposition provides a lower bound on the finite sample breakdown point of $DR_{p,\varepsilon}$, which highlights the robustness of the proposed distance as well as its dependence on $\varepsilon$.

**Proposition 3.7** (BREAKDOWN POINT). *For the halfspace depth function, for any $\beta \in [0, 1-\varepsilon]$ such that $\alpha(\beta, \hat{\mu}_n) < \alpha_{\max}(\hat{\mu}_n)$, it holds:*

$$BP(DR_{p,\varepsilon}, \mathcal{S}_n) \geq \begin{cases} \dfrac{\lceil n\alpha(1-\varepsilon, \hat{\mu}_n)/(1-\alpha(1-\varepsilon, \hat{\mu}_n)) \rceil}{n + \lceil n\alpha(1-\varepsilon, \hat{\mu}_n)/(1-\alpha(1-\varepsilon, \hat{\mu}_n)) \rceil} & if\, \alpha(1-\varepsilon, \hat{\mu}_n) \leq \frac{\alpha_{\max}(\hat{\mu}_n)}{1+\alpha_{\max}(\hat{\mu}_n)}, \\[4mm] \dfrac{\alpha_{\max}(\hat{\mu}_n)}{1 + \alpha_{\max}(\hat{\mu}_n)} & otherwise, \end{cases}$$

*where $\alpha_{\max}(\hat{\mu}_n) = \max_{x \in \mathbb{R}^d} HD_{\hat{\mu}_n}(x)$.*

Thus, at least a proportion $\alpha(1 - \varepsilon, \hat{\mu}_n)/(1 - \alpha(1 - \varepsilon, \hat{\mu}_n))$ of outliers must be added to break down $DR_{p,\varepsilon}$ when considering larger regions, while central regions are robust independently of $\varepsilon$. For two datasets, $DR_{p,\varepsilon}$ breaks down if depth regions for at least one of the datasets do. The breakdown point is then the minimum between the breakdown points of each dataset. However, the breakdown point considers the worst case, i.e. the supremum over all possible contaminations, and is often pessimistic. Indeed the proposed pseudo-metric can handle more outliers in certain cases, as experimentally illustrated in Section 5.1.

## 4 Efficient Approximate Computation

Exact computation of $DR_{p,\varepsilon}$ can appear time-consuming due to the high time complexity of the algorithms that calculate depth-trimmed regions (c.f. Liu & Zuo, 2014 and Liu et al., 2019a for projection and halfspace depths, respectively) rapidly growing with dimension. However, we design a universal approximate algorithm that achieves (log-) linear time complexity in $n$. Since properties (**D2**–**D4**) ensure that depth regions are convex bodies in $\mathbb{R}^d$, they can be characterized by their support functions defined by $h_{\mathcal{K}}(u) = \sup\{\langle x, u \rangle, \ x \in \mathcal{K}\}$ for any $u \in \mathbb{S}^{d-1}$ where $\mathcal{K}$ is a convex compact of $\mathbb{R}^d$. Following Schneider (1993), for two (convex) regions $D_\mu^{\alpha(\beta)}$ and $D_\nu^{\alpha(\beta)}$, the Hausdorff distance between them can be calculated as:

$$d_{\mathcal{H}}(D_\mu^{\alpha(\beta)}, D_\nu^{\alpha(\beta)}) = \sup_{u \in \mathbb{S}^{d-1}} \left| h_{D_\mu^{\alpha(\beta)}}(u) - h_{D_\nu^{\alpha(\beta)}}(u) \right|.$$

As we shall see in Section 5.1, mutual approximation of $h_{D^{\alpha(\beta)}}(u)$ by points from the sample and of sup by taking maximum over a finite set of directions allows for stable estimation quality. Recently, motivated by their numerous applications, many algorithms have been developed for the (exact and approximate) computation of data depths; see, e.g., Section 5 of Mosler & Mozharovskyi (2021) for a recent overview. Depths satisfying the projection property (which also include halfspace and projection depth, see Dyckerhoff (2004)) can be approximated by taking minimum over univariate depths; see e.g. Rousseeuw & Struyf (1998); Chen et al. (2013); Liu & Zuo (2014), Nagy et al. (2020) for theoretical guarantees, and Dyckerhoff et al. (2021) for an experimental validation.

**Empirical data.** Let $\mathbf{X}, \mathbf{Y}$ be two samples $\mathbf{X} = \{X_1, \ldots, X_n\}$ and $\mathbf{Y} = \{Y_1, \ldots, Y_m\}$ from $\mu, \nu$ such that $\hat{\mu}_n = (1/n) \sum_{i=1}^n \delta_{X_i}$ and $\hat{\nu}_n = (1/m) \sum_{i=1}^m \delta_{Y_m}$. When calculating approximated depth of sample points $D^{\mathbf{X}} \triangleq \{D(X_i, \hat{\mu}_n)\}_{i=1}^n$ (respectively $D^{\mathbf{Y}}$), a matrix $\mathbf{M}^{\mathbf{X}} \in \mathbb{R}^{n \times K}$ (respectively $\mathbf{M}^{\mathbf{Y}} \in \mathbb{R}^{m \times K}$) of projections of sample points on (a common) set of $K \in \mathbb{N}^*$ directions (with its element $\mathbf{M}_{i,k}^{\mathbf{X}} = \langle u_k, X_i \rangle$ for some $u_k \sim \mathcal{U}(\mathbb{S}^{d-1})$, where $\mathcal{U}(\cdot)$ is the uniform probability distribution) can be obtained as a side product. More precisely, $D^{\mathbf{X}}, D^{\mathbf{Y}}, \mathbf{M}^{\mathbf{X}}, \mathbf{M}^{\mathbf{Y}}$ are used in Algorithm 1, which implements the MC-approximation of the integral in (3). See Figure 1 for an illustration of the principle of this algorithm in practice.

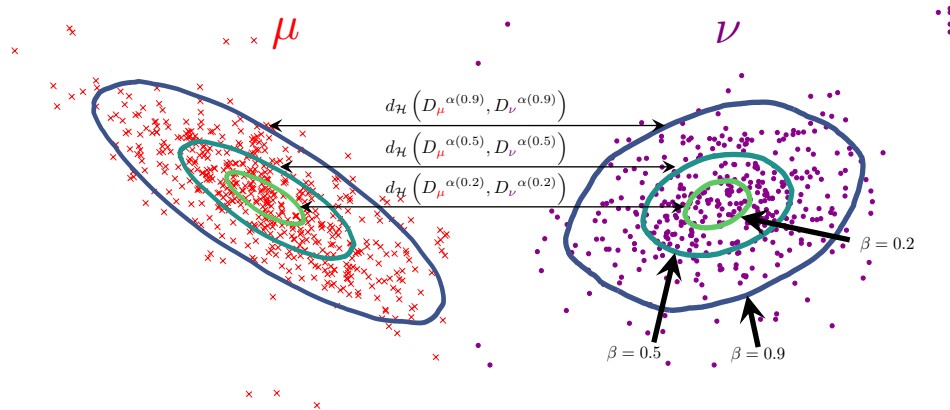

Figure 1: Illustration of the principle of the depth trimmed-regions based pseudo-metric with $n_\alpha = 3$ and $\beta = \{0.2, 0.5, 0.9\}$.

Particular cases of approximation algorithms for the halfspace depth and the projection depth are recalled in Section C in the Appendix. Time complexity of Algorithm 1 is $O\big(K(\Omega.(n \vee m, d) \vee n_\alpha(n \vee m))\big)$, where $\Omega.(\cdot, \cdot)$ stands for the complete complexity of computing univariate depths—in projections on $u$—for all points of the sample. As a byproduct, projections on $u$ can be saved to be reused after for the approximation of $h_{D^{\alpha(\beta)}}(u)$. For the halfspace depth $\Omega_{hsp}(n, d) = O\big(n(d \vee \log n)\big)$ composed of projection of the data onto $u$, ordering them, and passing to record the depths, see Mozharovskyi et al. (2015). For the projection depth, $\Omega_{prj}(n, d) = O(nd)$, where after projecting the data onto $u$, univariate median and MAD can be computed with complexity $O(n)$, see Liu & Zuo (2014)). In comparison with popular distances, fixing $n = m$, the Wasserstein distance is of order $O(n^2(d \vee n))$ with approximations in $O(n^2 d)$ for Sinkhorn (Cuturi et al., 2013) and in $O(Kn(d \vee \log(n)))$ for the Sliced-Wasserstein distance (Rabin et al., 2012); the MMD (Gretton et al., 2007) is of order $O(n^2 d)$. For example, the computational complexity of $DR_{p,\varepsilon}$ with the projection depth is only of $O(Kn(d \vee n_\alpha))$ and thus competes with the fastest (max) sliced-Wasserstein distance.

---

**Algorithm 1** Approximation of $DR_{p,\varepsilon}$

---

*Initialization:* $\mathbf{X}, \mathbf{Y}, n_\alpha, K$

 1: $H = 0$; compute $D^{\mathbf{X}}, D^{\mathbf{Y}}, \mathbf{M}^{\mathbf{X}}, \mathbf{M}^{\mathbf{Y}}$
 2: **for** $\ell = 1, \ldots, n_\alpha$ **do**
 3:     Draw $\beta_\ell \sim \mathcal{U}([0, 1 - \varepsilon])$
 4:     Compute $\hat{\alpha}_\ell(\cdot) := \hat{\alpha}(\beta_\ell, \cdot)$
 5:     Determine points inside $\alpha_\ell(\cdot)$-regions:
     $\mathcal{I}_\ell^{\mathbf{X}} = \{i : D_i^{\mathbf{X}} > \hat{\alpha}_\ell(\mathbf{X})\}; \ \mathcal{I}_\ell^{\mathbf{Y}} = \{j : D_j^{\mathbf{Y}} > \hat{\alpha}_\ell(\mathbf{Y})\}$
 6:     **for** $k = 1, \ldots, K$ **do**
 7:         Compute approximation of support functions: $h_k^{\mathbf{X}} = \max \mathbf{M}_{\mathcal{I}_\ell^{\mathbf{X}}, k}^{\mathbf{X}}; \ h_k^{\mathbf{Y}} = \max \mathbf{M}_{\mathcal{I}_\ell^{\mathbf{Y}}, k}^{\mathbf{Y}}$
 8:     **end for**
 9:     Increase cumulative Hausdorff distance:
     $H \mathrel{+}= \max_{k \le K} |h_k^{\mathbf{X}} - h_k^{\mathbf{Y}}|^p$
10: **end for**
    **Output**: $\widehat{DR}_{p,\varepsilon} = (H/n_\alpha)^{1/p}$

---

## 5 Numerical Experiments

In this section, we first investigate different properties of the proposed pseudo-metric such as the convergence rates of the pseudo-metric estimator w.r.t. the sample size, the quality of the approximation introduced in Section 4 and its dependency on the number of projections. Further, we present two studies on the robustness of the proposed pseudo-metric $DR_{p,\varepsilon}$ to outliers. Finally, we show the performance of this pseudo-metric on two machine learning tasks, clustering and automatic evaluation of neural language generation. Where applicable, we include state-of-the-art methods for comparison.

### 5.1 Statistical Convergence, Approximation and Robustness

This part describes the behavior of the proposed pseudo-metric through different perspectives. On synthetic datasets, we investigate the statistical convergence rates of the empirical version of $DR_{p,\varepsilon}$ to the population one. We assess the Monte Carlo approximation proposed in Section 4 and compare it to the Sliced Wasserstein distance. Finally, we highlight how $DR_{p,\varepsilon}$ behaves under the presence of outliers using two different settings. Due to space limitations, experiments on the influence of the parameters $n_\alpha$ and $\varepsilon$ are deferred to the Appendix section.

**Empirical analysis of statistical rates.** Deriving theoretical finite-sample analysis may appear to be challenging for the proposed pseudo-metric. Thus, we numerically investigate the statistical convergence speed of $DR_{2,\varepsilon}$. To that end, we simulate two samples $\mathbf{X}$ and $\mathbf{Y}$ from two standard Gaussian distributions in dimension two with varying sample sizes from $n = 10$ to $n = 10000$, see Section D.3 for additional experiments with $d \in \{5, 10\}$. We compute the $DR_{2,\varepsilon}$ between $\mathbf{X}$ and $\mathbf{Y}$ with $n_\alpha \in \{5, 20, 100\}$ using the halfspace and

the projection depths. Our proposed metric is computed with a high number of directions $K = 25000$ to isolate the statistical error. We report the estimation error averaged over ten runs in Figure 2 (log-log scale), that is, the value of the pseudo-metric itself, the true value of $DR_{2,\varepsilon}$ being equal to zero. When the Monte Carlo approximation error influenced by the parameter $n_\alpha$ is negligible ($n_\alpha = 100$), Figure 2 suggests that the statistical rates should be in $O(n^{-1/4})$. Furthermore, Figure 7 indicates a rates of order $O(n^{-0.8/4})$ and Figure 8 of order $O(n^{-0.6/4})$. These observations suggest a slow rate that depends on the dimension $d$ of the data. However, the approximation error being negligible due to the $K = 25000$ sampled directions, the statistical rates seem to depend only linearly on the dimension. Looking at the error values for $n = 10000$ for $d = 2, 5, 10$, it increases by a factor of two, such as the dimension.

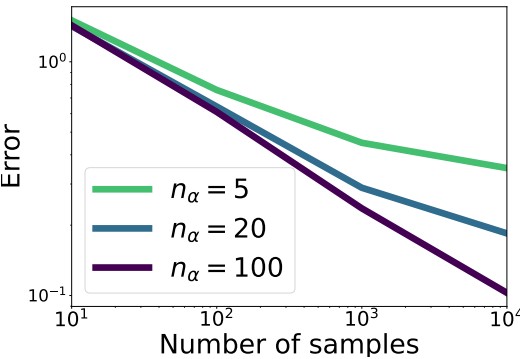
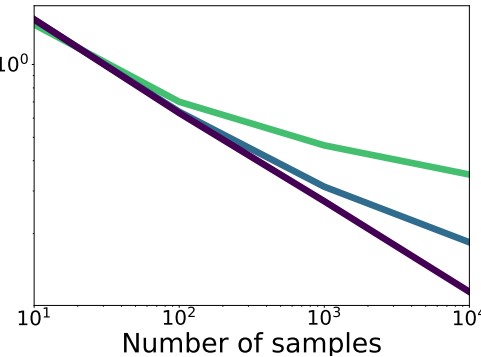

Figure 2: Empirical analysis of statistical convergence rates. Resulting error of the proposed pseudo-metric when increasing the sample size using the projection depth (left) and the halfspace depth (right) for various $n_\alpha$ parameters.

**Approximation error in terms of the number of projections.** Proposition 3.6 allows to derive a closed form expression for $DR_{2,\varepsilon}(\mu, \nu)$ when $\mu, \nu$ are Gaussian distributions with the same variance-covariance matrix. In order to investigate the quality of the approximation on light-tailed and heavy-tailed distributions, we focus on computing $DR_{p,\varepsilon}$ with $p = 2$, $\varepsilon = 0.3$, $n_\alpha = 20$ and using the halfspace depth for varying number of random projections $K$ between a sample of 1000 points stemming from $\mu \sim \mathcal{N}(\mathbf{0}_d, I_d)$ for $d = 5$ and two different samples. These two samples are constructed from 1000 observations stemming from *Gaussian* and symmetrical *Cauchy* distributions, both with a center equal to $\mathbf{7}_d$. Comparison with the approximation of max Sliced-Wasserstein (max-SW; see e.g. Kolouri et al., 2019), which shares the same closed-form as $DR_{2,\varepsilon}$, is also provided. Denoting by max-$\widehat{\text{SW}}$ the Monte-Carlo approximation of the max-SW, the relative approximation errors, i.e., $(\widehat{DR}_{p,\varepsilon} - ||\mathbf{7}_d||_2)/||\mathbf{7}_d||_2$ and $(\text{max-}\widehat{\text{SW}} - ||\mathbf{7}_d||_2)/||\mathbf{7}_d||_2$, are computed investigating both the quality of the approximation and the robustness of these discrepancy measures. Results that report the averaged approximation error and the 25-75% empirical quantile intervals are depicted in Figure 3. They show that $DR_{p,\varepsilon}$ possesses the same behavior as max-SW when considering *Gaussians* while it behaves advantageously for *Cauchy* distribution. Computation times are depicted in Figure 4, highlighting a constant-multiple improvement compared to the max-SW, which is already computationally fast.

**Robustness to outliers.** We analyze the robustness of $DR_{p,\varepsilon}$ by measuring its ability to overcome outliers (its robustness regarding the influence of the parameter $\varepsilon$ are given in the Section D.4 in the Appendix). In this benchmark, we naturally include existing robust extensions of the Wasserstein distance: Subspace Robust Wasserstein (SRW; Paty & Cuturi, 2019) searching for a maximal distance on lower-dimensional subspaces, ROBOT (Mukherjee et al., 2020) and RUOT (Balaji et al., 2020) being robust modifications of the unbalanced optimal transport (Chizat et al., 2018). Medians-of-Means Wasserstein (MoMW; Staerman et al., 2021a) that replaces the empirical means in the Kantorivich duality formulae by the robust mean estimator MoM (see e.g. Lecué & Lerasle, 2020; Laforgue et al., 2021), is not employed due to high computational burden. Further, for completeness, we add the standard Wasserstein distance (W) and its approximation, the Sliced-Wasserstein (Sliced-W; Rabin et al., 2012) distance, the trimmed Sliced-Wasserstein (TSW; Manole

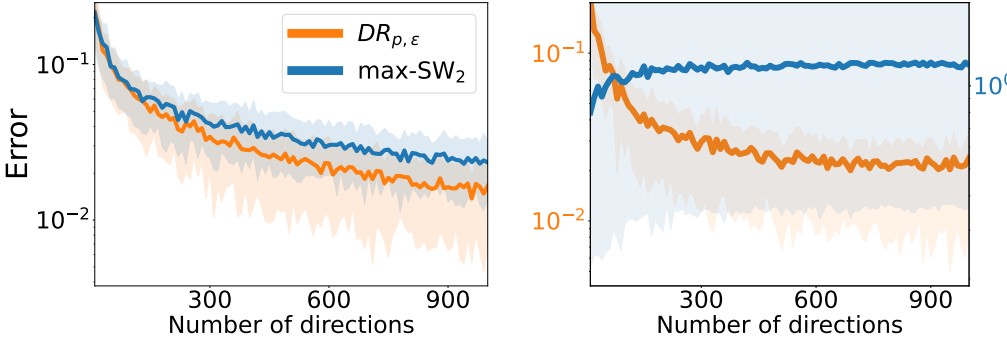

Figure 3: Relative approximation error (averaged over 100 runs) of $DR_{p,\varepsilon}$ and the max Sliced-Wassserstein for *Gaussian* (left) and *Cauchy* (right) sample with dimension $d = 5$ for differing numbers of approximating directions.

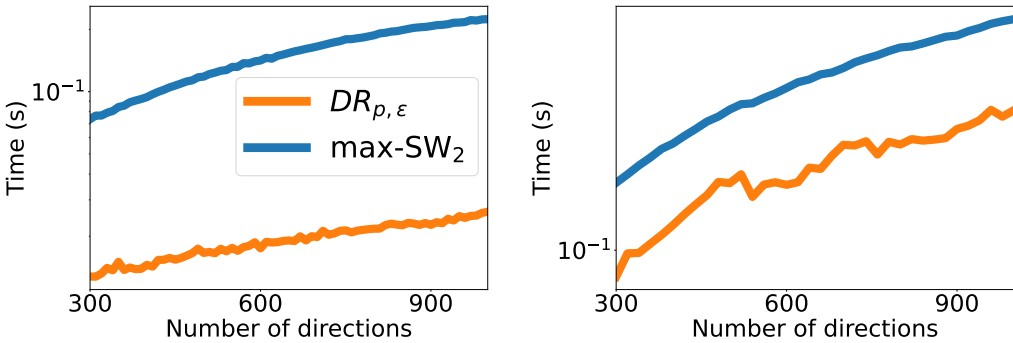

Figure 4: Computation time (averaged over 100 runs) of $DR_{p,\varepsilon}$ and the max Sliced-Wassserstein for *Gaussian* with $n = 100$, $d = 5$ (left) and $n = 1000$, $d = 50$ (right) for differing numbers of approximating directions.

et al., 2022) with the same number of projections ($K = 1000$) as $DR_{p,\varepsilon}$. Since the scales of the compared methods differ, *relative error* is used as a performance metric, i.e., the ratio of the absolute difference of the computed distance with and without anomalies divided by the latter. Two settings for a pair of distributions are addressed: (a) *Fragmented hypercube* precedently studied in Paty & Cuturi (2019), where the source distribution is uniform in the hypercube $[-1, 1]^2$ and the target distribution is transformed from the source via the map $T : x \mapsto x + 2\text{sign}(x)$ where $sign(.)$ is taken element-wisely. Outliers are drawn uniformly from $[-4, 4]^2$. (b) Two multivariate standard *Gaussian* distributions, one shifted by $\mathbf{10}_2$, with outliers drawn uniformly from $[-10, 20]^2$. Our analysis is conducted over 500 sampled points from the distributions described above.

To investigate the robustness of $DR_{p,\varepsilon}$, we consider the following value of $\varepsilon$: 0.3 computed with the projection depth. We set the same trimming value for TSW. Thus, data depths are computed on source and target distributions such that 30% of data with lower depth values w.r.t. each distribution are not used in computation of $DR_{p,0.3}$, respectively. Figure 5, which plots the relative error depending on the portion of outliers varying up to 30%, illustrates advantageous behavior of $DR_{p,\varepsilon}$ for reasonable (starting with $\approx 2.5\%$) contamination.

## 5.2   Machine Learning Applications

This part presents two machine learning applications, clustering applied to images and automatic evaluation of natural language generation. On a real image dataset extracted from Fashion-MNIST where images are seen as bags of pixels, we evaluate the robustness of spectral clustering based on $DR_{p,\varepsilon}$. Further, we analyze

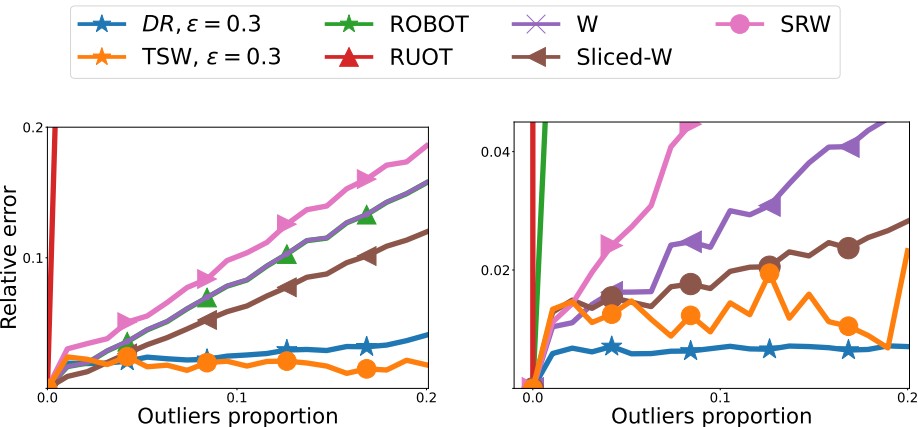

Figure 5: Relative error (averaged over 100 runs) of different distances for increasing outliers proportion on *fragmented hypercube* (left) and *Gaussian* (right) data.

the relevance of using $DR_{p,\varepsilon}$ as an evaluation metric in natural language generation to compare the empirical distributions of words of a pair of texts.

**(Robust) Clustering on bags of pixels.** We demonstrate the relevance of the proposed pseudo-metric through an application to (robust) clustering. To that end, we perform spectral clustering (Shi & Malik, 2000) on two datasets derived from Fashion-MNIST (FM). Each grayscale image is seen as a bag of pixels (Jebara, 2003), i.e. as an empirical probability distribution over a 3-dimensional space (the two first dimensions indicate the pixel position and the third one, its intensity). The first dataset (FM) is constructed by taking the 100 first images in each class of the Fashion-MNIST dataset. The second dataset (Cont. FM), considered contaminated, is designed by introducing white patches on the left corner of 50 images drawn uniformly in the first dataset, which yields 5% of contamination. We benchmark $DR_{p,\varepsilon}$ (using the projection depth) setting $p = 2$ and $\varepsilon = 0.1$ with the Wasserstein (W), the Sliced-Wasserstein (Sliced-W) and the Maximum Mean Discrepancy (MMD; Gretton et al., 2007) distances. $DR_{p,\varepsilon}$ and the Sliced-Wasserstein are approximated by Monte-Carlo using 100 directions while the MMD distance is computed using a Gaussian kernel with a bandwidth equal to 1. As a baseline method, spectral clustering is also applied to images considered as vectors using Euclidean distance. Standard parameters of the `scikit-learn` spectral clustering implementation are employed with a number of clusters fixed to 10. Performances of the benchmarked metrics are assessed by measuring the normalized mutual information (NMI; Shannon, 1948) and the adjusted rank index (ARI; Hubert & Arabie, 1985), which are standard clustering evaluation measures when the ground truth class labels are available. Results presented in Table 1 show that for both cases, i.e. with or without contamination, spectral clustering based on $DR_{p,\varepsilon}$ outperforms spectral clustering based on the other metrics.

| | FM | | Cont. FM | |
|---|---|---|---|---|
| | NMI | ARI | NMI | ARI |
| $DR_{p,\varepsilon}$ | **0.58** | **0.43** | **0.55** | **0.42** |
| W | 0.50 | 0.35 | 0.48 | 0.30 |
| Sliced-W | 0.55 | 0.39 | 0.47 | 0.33 |
| MMD | 0.54 | 0.37 | 0.50 | 0.36 |
| Euclidean | 0.50 | 0.32 | 0.48 | 0.30 |

Table 1: Spectral clustering performances.

**Automatic evaluation of natural language generation (NLG).** Collecting human annotations to evaluate NLG systems is both expensive and time-consuming. Thus, automatically assessing the similarity

| | Correctness | | | Data Coverage | | | Relevance | | |
|---|---|---|---|---|---|---|---|---|---|
| | $r$ | $\tau$ | $\rho$ | $r$ | $\tau$ | $\rho$ | $r$ | $\tau$ | $\rho$ |
| $DR_{p,\varepsilon}$ | **89.4** | **80.0** | **92.6** | **84.2** | 58.3 | 72.3 | **86.2** | 62.7 | 72.9 |
| W | 86.2 | 73.0 | 86.7 | 80.4 | 45.3 | 62.3 | 83.8 | 51.3 | 67.6 |
| Sliced-W | 86.1 | 73.0 | 85.8 | 80.9 | 45.5 | 60.0 | 82.0 | 51.3 | 68.2 |
| MMD | 25.4 | 71.7 | 8.3 | 19.1 | 45.3 | 10.0 | 26.1 | 51.3 | 15.0 |
| BertS | 85.5 | 73.3 | 83.4 | 74.7 | 53.3 | 68.2 | 83.3 | 65.0 | **79.4** |
| MoverS | 84.1 | 73.3 | 84.1 | 78.7 | 53.3 | 66.2 | 82.1 | 65.0 | 77.4 |
| BLEU | 77.6 | 60.0 | 66.3 | 55.7 | 36.6 | 50.2 | 63.0 | 51.6 | 65.2 |
| ROUGE | 80.6 | 65.0 | 65.0 | 76.5 | **60.3** | **76.3** | 64.3 | 56.7 | 69.2 |
| MET. | 86.5 | 70.0 | 66.3 | 77.3 | 46.6 | 50.2 | 82.1 | 58.6 | 65.2 |
| TER | 79.6 | 58.0 | 78.3 | 69.7 | 38.0 | 58.2 | 75.0 | **77.6** | 70.2 |

Table 2: Absolute correlation at the system level with three human judgment criteria. The best overall results are indicated in bold, best results in their group are underlined.

between two texts is highly interesting for the NLP community (Specia et al., 2010). This task aims to build an evaluation metric that achieves a high correlation with the score given by a human annotator. String-based metrics (i.e. that compare the string representations of texts) such as BLEU (Papineni et al., 2002), METEOR (MET.; Banerjee & Lavie, 2005), ROUGE (Lin, 2004), TER (Snover et al., 2006), have been outperformed in many tasks by embedding-based metrics, i.e., that rely on continuous representations (Devlin et al., 2019). Embedding-based metrics, e.g BertScore (BertS; Zhang et al., 2019) and MoverScore (MoverS; Zhao et al., 2019) that are now the state-of-the-art domain, compare input and reference texts both represented as probability distributions and are both constructed similarly. The first step relies on a deep contextualized encoder (BERT in our case, see Devlin et al., 2019) that maps texts into elements of a finite-dimensional space. Each text corresponds to a collection of words, where each word is represented by an element in $\mathbb{R}^d$, where $d$ is fixed by the encoder. The second step involves using a function that measures the similarity between the embedded texts.

We follow previous BERT-based metrics and evaluate performances of $DR_{p,\varepsilon}$ (with $p = 2$, $\varepsilon = 0.01$ and using the AI-IRW depth (Staerman et al., 2021b)) on two different NLG tasks namely: data2text generation (using the WebNLG 2020 dataset (Ferreira et al., 2020)) and summarization. For the sake of place, summarization results and additional experimental details are reported in Section E in the Appendix. For WebNLG, we follow standard methods to assess the performance of NLG metrics (see e.g. Zhao et al., 2019). We compute the correlation with the following annotation scores: *correctness*, *data coverage*, and *relevance*. We report in Table 2 correlation results on the WebNLG task using Pearson ($r$), Spearman ($\rho$) and Kendall ($\tau$) correlation coefficients. When performing a fair comparison between metrics, i.e. when $DR_{p,\varepsilon}$, W, Sliced-W, MMD are directly used on the output of BERT, we observe that $DR_{p,\varepsilon}$ achieves the best results on all configurations. It is worth noting that $DR_{p,\varepsilon}$ also compares favorably against existing state-of-the-art NLG methods in many different scenarios and shows promising results.

## 6 Discussion

Leveraging the notion of statistical data depth function, a novel pseudo-metric between multivariate probability distributions—that meets the aforementioned requirements—was introduced. The developed framework exhibits inherent versatility due to numerous data depth variants. The linear approximation algorithm and the robustness property make $DR_{p,\varepsilon}$ a promising tool for a large spectrum of applications beyond

clustering and NLG, e.g. in generative adversarial networks (GANs) or information retrieval. Moreover, recent works extending the notion of data depth to further types of data such as functional and time-series data (Nieto-Reyes & Battey, 2016; Gijbels & Nagy, 2017), directional (or spherical) data (Ley et al., 2014), random matrices (Paindaveine & Van Bever, 2018), curves (or paths) data (Lafaye et al., 2020), and random sets (Cascos et al., 2021) shall allow for the use of the proposed pseudo-metric for a wide range of applications.

## Acknowledgments

The authors thank the Jean Zay supercomputer operated by GENCI IDRIS with the compute grant 2023-AD011014668R1 and Adastra with the grant AD010614770, where the NLP experiments have been done.

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

## Appendix

This Appendix is organized as follows:

- Appendix A contains additional notations, preliminary results and additional information about the proposed pseudo-metric.

- Appendix B contains the proofs of the propositions/theorems provided in the paper.

- Appendix C contains approximation algorithms to compute halfspace/projection/AI-IRW depth.

- Appendix D contains additional synthetic experiments.

- Appendix E contains details on experimental settings of NLP applications.

# A   Preliminary Results

First, we introduce additional notations and recall some lemmas, used in the subsequent proofs.

## A.1   Hausdorff Distance

The Hausdorff distance between two bounded subspaces $\mathcal{K}_1, \mathcal{K}_2$ of $\mathbb{R}^d$ is defined as:

$$d_{\mathcal{H}}(\mathcal{K}_1, \mathcal{K}_2) = \max \left\{ \sup_{x \in \mathcal{K}_1} \inf_{y \in \mathcal{K}_2} ||x - y||, \ \sup_{y \in \mathcal{K}_2} \inf_{x \in \mathcal{K}_1} ||x - y|| \right\}.$$

Furthermore, if $\mathcal{K}_1$ and $\mathcal{K}_2$ are convex bodies, i.e. non empty compact convex sets, the Hausdorff distance can be reformulated with support functions of $\mathcal{K}_1, \mathcal{K}_2$:

$$d_{\mathcal{H}}(\mathcal{K}_1, \mathcal{K}_2) = \sup_{u \in \mathbb{S}^{d-1}} \left| h_{\mathcal{K}_1}(u) - h_{\mathcal{K}_2}(u) \right|,$$

where $h_{\mathcal{K}_1}(u) = \sup\{\langle u, x \rangle, \ x \in \mathcal{K}_1\}$.

## A.2   Quantile Regions

Let $u \in \mathbb{S}^{d-1}$ and $X \sim \mu$ where $\mu \in \mathcal{M}_1(\mathcal{X})$ with $\mathcal{X} \subset \mathbb{R}^d$. We define the $(1 - \beta)$ directional quantile of a distribution $\mu$ in the direction $u$ as:

$$q_{\mu,u}^{1-\beta} = \inf \left\{ t \in \mathbb{R} : \ \mathbb{P}\left( \langle u, X \rangle \leq t \right) \geq 1 - \beta \right\},$$

and the upper $(1 - \beta)$ quantile set of $\mu$:

$$Q_{\mu}^{1-\beta} = \left\{ x \in \mathbb{R}^d : \ \langle u, x \rangle \leq q_{\mu,u}^{1-\beta}, \ \ \forall \, u \in \mathbb{S}^{d-1} \right\}.$$

## A.3   Auxiliary Results

We now recall useful results, so as to characterize the halfspace depth regions.

**Lemma A.1** (Brunel, 2019, Lemma 1). *Let $\mu \in \mathcal{M}_1(\mathcal{X})$, for any $\beta \in (0, 1)$, it holds: $D_{\mu}^{\beta} = Q_{\mu}^{1-\beta}$.*

**Lemma A.2** (Brunel, 2019, Proposition 1). *Let $\mu \in \mathcal{M}_1(\mathcal{X})$ with a $(1 - \beta)$ directional quantile $q_{\mu,u}^{1-\beta}$ for any $u \in \mathbb{S}^{d-1}$. Assume that $u \mapsto q_{\mu,u}^{1-\beta}$ are sublinear, i.e., $q_{\mu,u+\lambda v}^{1-\beta} \leq q_{\mu,u}^{1-\beta} + \lambda \, q_{\mu,v}^{1-\beta}, \ \ \forall \, \lambda > 0$. Then for any $u \in \mathbb{S}^{d-1}$, it holds $h_{Q_{\mu,u}^{1-\beta}}(u) = q_{\mu,u}^{1-\beta}$.*

**Lemma A.3.** *Let $d = 1$ and $X^1 \sim \mu_1$, $Y^1 \sim \nu_1$ be two random variables where $\mu_1, \nu_1$ are univariate probability distributions. Denoting by $F_{X^1}^{-1}$ the quantile function of $X^1$, then the depth-trimmed region based pseudo-metric (associated with the halfspace depth) is defined as*

$$DR_{p,\varepsilon}^p(\mu_1, \nu_1) = 2 \int_{\varepsilon/2}^{1/2} \max \left\{ |F_{X^1}^{-1}(q) - F_{Y^1}^{-1}(q)|^p, \ |F_{X^1}^{-1}(1 - q) - F_{Y^1}^{-1}(1 - q)|^p \right\} \mathrm{d}q.$$

*Proof.* In dimension one, the halfspace depth of any $t \in \mathbb{R}$ w.r.t. $\mu_1$ and $\nu_1$ boils down to

$$D(t, \mu_1) = \min \left\{ F_{X^1}(t), 1 - F_{X^1}(t) \right\} \quad \text{and} \quad D(t, \nu_1) = \min \left\{ F_{Y^1}(t), 1 - F_{Y^1}(t) \right\},$$

and for any $\gamma \in [0, 1]$, its upper-level sets to intervals

$$D_{\mu_1}^{\gamma} = [F_{X^1}^{-1}(\gamma), \ F_{X^1}^{-1}(1 - \gamma)] \quad \text{and} \quad D_{\nu_1}^{\gamma} = [F_{Y^1}^{-1}(\gamma), \ F_{Y^1}^{-1}(1 - \gamma)]. \tag{5}$$

Now, the quantile function $\alpha(\beta, .)$ can be explicitly derived as function of $\beta \in [0, 1]$:

$$
\begin{aligned}
\alpha(\beta, \mu_1) &= \sup\left\{\gamma \in [0, 1]: \ \mu_1\left(\left[F_{X^1}^{-1}(\gamma), \ F_{X^1}^{-1}(1 - \gamma)\right]\right) \geq \beta\right\} \\
&= \sup\left\{\gamma \in [0, 1]: \ 1 - 2\gamma \geq \beta\right\} \\
&= \frac{1 - \beta}{2}.
\end{aligned}
$$

Following the same reasoning, it holds $\alpha(\beta, \nu_1) = \frac{1-\beta}{2}$. Further, by change of variables

$$
\int_0^{1-\varepsilon} d_{\mathcal{H}}\left(D_{\mu_1}^{(1-\beta)/2}, D_{\nu_1}^{(1-\beta)/2}\right)^p \, \mathrm{d}\beta = 2\int_{\varepsilon/2}^{1/2} d_{\mathcal{H}}\left(D_{\mu_1}^q, D_{\nu_1}^q\right)^p \, \mathrm{d}q.
$$

Combining (5) and the Hausdorff distance definition recalled in Section A.1 lead to the result.

$\square$

## A.4 Additional information

This part provides additional information and remarks about the proposed pseudo-metric. We also summarize properties of $DR_{p,\varepsilon}$ in Table 3 w.r.t. different depth functions used in the paper.

First, in some cases of convex data depth, the pseudo-metric could define a distance. $DR_{p,\varepsilon}$ is a distance if and only if the upper-level sets of the chosen data depth fully characterize probability distributions. To our knowledge, It has been proved only for the Halfspace depth under mild assumptions (Hassairi & Regaieg, 2008; Nagy et al., 2019) and the Zonoid depth assuming the first moment on the distribution (Mosler, 2002).

**Remark A.4** (FLEXIBILITY.)**.** *One of the main benefits of our pseudo-metric is its flexibility. Our general definition allows the use of any depth function, see e.g. Mosler & Mozharovskyi (2022) for a review of the main depths, at the price of choosing one that is relevant for the underlying data.*

**Remark A.5** (ROBUSTNESS.)**.** *The trimming improves the robustness of $DR_{p,\varepsilon}$ and sliced-Wasserstein. While it is not the only source of robustness in $DR_{p,\varepsilon}$, the Wasserstein metric is generally known to be non-robust (Mukherjee et al., 2021; Balaji et al., 2020). Indeed, the robustness in $DR_{p,\varepsilon}$ also comes from the robustness of the chosen depth and the trimming step may not be helpful in specific situations.*

**Remark A.6** (TRIMMING COMPARISON WITH TSW.)**.** *The trimming occurs in different spaces for the trimmed sliced Wasserstein and $DR_{p,\varepsilon}$. Our pseudo-metric trimming relies on peeling the larger quantile regions (depth regions) that directly consider the multivariate data's structure. In contrast, the trimming in SW occurs at the projection level. Since, in practice, the projections are chosen uniformly on the unit sphere, this trimming does not consider the correlation of the data, for example. It may remove non-outlier points of the data.*

## B  Technical Proofs

We now prove the main results stated in the paper.

### B.1  Proof of Proposition 3.4

For any $0 \leq \beta \leq 1 - \varepsilon$ with $\varepsilon \in (0, 1]$, and any $\mu \in \mathcal{M}_1(\mathcal{X})$, $\nu \in \mathcal{M}_1(\mathcal{Y})$, $D_\mu^{\alpha(\beta)}, D_\nu^{\alpha(\beta)}$ are non-empty compact subsets of $\mathbb{R}^d$ due to the properties (**D2-D3**). The Hausdorff distance $d_{\mathcal{H}}$, recalled in Section A.1, is known to be a distance on the space of non-empty compact sets which implies that $DR_{p,\varepsilon}$ satisfies positivity, symmetry and the triangle inequality (thanks to Minkowski inequality). If $\mu = \nu$ then $D_\mu^{\alpha(\beta)} = D_\nu^{\alpha(\beta)}, \quad \forall \beta \in [0, 1-\varepsilon]$ which

|  | Halfspace | Projection | IRW | AI-IRW |
|---|---|---|---|---|
| Pseudo-metric | ✓ | ✓ | ✓ | ✓ |
| Isometry invariance | ✓ | ✓ | ✗ | ✓ |
| Fast approximation with support vector | ✓ | ✓ | ✗ | ✓ |
| Depth computation | $O(Kn(d \vee \log n))$ | $O(ndK)$ | $O(ndK)$ | $O(d^3 + ndK)$ |
| Robust depth regions | ✗ | ✓ | ✗ | ✓ |

Table 3: Properties satisfied by $DR_{p,\varepsilon}$ associated with the halfspace, projection and affine-invariant integrated rank-weighted depths.

leads to $DR_{p,\varepsilon}(\mu, \nu) = 0$. The reverse is not true. $DR_{p,\varepsilon}(\mu, \nu) = 0$ implies $D_\mu^{\alpha(\beta)} = D_\nu^{\alpha(\beta)}$, $\quad \forall \beta \in [0, 1-\varepsilon]$ that not leads to $\mu = \nu$. Indeed, convex depth regions do not characterize probability distributions in general (see Nagy, 2019 for the halfspace depth) that would be the first step in order to prove the previous entailment.

## B.2 Proof of Proposition 3.5

Let $A \in \mathbb{R}^{d \times d}$ be a non-singular matrix and $b \in \mathbb{R}^d$ such that $g : x \mapsto Ax + b$. Then, it holds:

$$
\begin{aligned}
DR_{p,\varepsilon}^p(g_\sharp \mu, g_\sharp \nu) &= \int_0^{1-\varepsilon} \left[ d_{\mathcal{H}}(D_{g_\sharp \mu}^{\alpha(\beta)}, D_{g_\sharp \nu}^{\alpha(\beta)}) \right]^p \, \mathrm{d}\beta \\
&\overset{(i)}{=} \int_0^{1-\varepsilon} \left[ d_{\mathcal{H}}(AD_\mu^{\alpha(\beta)} + b, AD_\nu^{\alpha(\beta)} + b) \right]^p \, \mathrm{d}\beta,
\end{aligned}
\tag{6}
$$

where $(i)$ holds because any data depth satisfies (**D1**) by definition. Furthermore,

$$
\begin{aligned}
d_{\mathcal{H}}(AD_\mu^{\alpha(\beta)} + b, AD_\nu^{\alpha(\beta)} + b) &= \max \left\{ \sup_{x \in D_\mu^{\alpha(\beta)}} \inf_{y \in D_\nu^{\alpha(\beta)}} ||Ax - Ay||, \sup_{y \in D_\nu^{\alpha(\beta)}} \inf_{x \in D_\mu^{\alpha(\beta)}} ||Ax - Ay|| \right\} \\
&\overset{(ii)}{=} \max \left\{ \sup_{x \in D_\mu^{\alpha(\beta)}} \inf_{y \in D_\nu^{\alpha(\beta)}} ||x - y||, \sup_{y \in D_\nu^{\alpha(\beta)}} \inf_{x \in D_\mu^{\alpha(\beta)}} ||x - y|| \right\} \\
&= d_{\mathcal{H}}(D_\mu^{\alpha(\beta)}, D_\nu^{\alpha(\beta)}),
\end{aligned}
$$

where $(ii)$ holds by virtue of hypothesis $AA^\top = I_d$. Replacing it in (6) yields the desired results.

## B.3 Proof of Proposition 3.6

**First assertion.** Denote $Z_1, Z_2$ two random variables following $\mu^*, \nu^*$ respectively. Assume that $X, Y, Z_1, Z_2$ are defined on the probability space $(\Omega, \mathcal{A}, \mathbb{P})$. For any $x \in \mathbb{R}^d$ and $\beta \in [0, 1-\varepsilon]$,

$$
\begin{aligned}
x \in D_\mu^{\alpha(\beta)} \iff HD_\mu(x) \geq \alpha(\beta) &\iff \forall u \in \mathbb{S}^{d-1}, \quad \mathbb{P}(\langle u, X \rangle \leq \langle u, x \rangle) \geq \alpha(\beta) \\
&\iff \forall u \in \mathbb{S}^{d-1}, \quad \mathbb{P}(\langle u, Z_1 + \mathbf{m}_1 \rangle \leq \langle u, x \rangle) \geq \alpha(\beta) \\
&\iff \forall u \in \mathbb{S}^{d-1}, \quad \mathbb{P}(\langle u, Z_1 \rangle \leq \langle u, x - \mathbf{m}_1 \rangle) \geq \alpha(\beta) \\
&\iff x - \mathbf{m}_1 \in D_{\mu^*}^{\alpha(\beta)}.
\end{aligned}
$$

The same reasoning holds for $\nu$ and $\nu^*$. Following this, for any $\beta \in [0, 1-\varepsilon]$ and $u \in \mathbb{S}^{d-1}$, it holds:

$$h_{D_\mu^{\alpha(\beta)}}(u) = h_{D_{\mu^*}^{\alpha(\beta)}}(u) - \langle u, \mathbf{m}_1 \rangle \qquad \text{and} \qquad h_{D_\nu^{\alpha(\beta)}}(u) = h_{D_{\nu^*}^{\alpha(\beta)}}(u) - \langle u, \mathbf{m}_2 \rangle.$$

Thus it holds:

$$
\begin{aligned}
DR_{2,\varepsilon}^2(\mu, \nu) &= \int_0^{1-\varepsilon} \sup_{u \in \mathbb{S}^{d-1}} \left| h_{D_{\mu^*}^{\alpha(\beta)}}(u) - \langle u, \mathbf{m}_1 \rangle - h_{D_{\nu^*}^{\alpha(\beta)}}(u) + \langle u, \mathbf{m}_2 \rangle \right|^2 \, \mathrm{d}\beta \\
&\leq \sup_{u \in \mathbb{S}^{d-1}} \left| \langle u, \mathbf{m}_1 - \mathbf{m}_2 \rangle \right|^2 + \int_0^{1-\varepsilon} \sup_{u \in \mathbb{S}^{d-1}} \left| h_{D_{\mu^*}^{\alpha(\beta)}}(u) - h_{D_{\nu^*}^{\alpha(\beta)}}(u) \right|^2 \, \mathrm{d}\beta \\
&\quad + 2 \sup_{u \in \mathbb{S}^{d-1}} \left| \langle u, \mathbf{m}_1 - \mathbf{m}_2 \rangle \right| \int_0^{1-\varepsilon} \sup_{u \in \mathbb{S}^{d-1}} \left| h_{D_{\mu^*}^{\alpha(\beta)}}(u) - h_{D_{\nu^*}^{\alpha(\beta)}}(u) \right| \, \mathrm{d}\beta \\
&= ||\mathbf{m}_1 - \mathbf{m}_2||^2 + DR_{2,\varepsilon}^2(\mu^*, \nu^*) + 2||\mathbf{m}_1 - \mathbf{m}_2|| DR_{1,\varepsilon}(\mu^*, \nu^*).
\end{aligned}
\tag{7}
$$

On the other side, we have:

$$
\begin{aligned}
DR_{2,\varepsilon}^2(\mu, \nu) &\geq \sup_{u \in \mathbb{S}^{d-1}} \left| \langle u, \mathbf{m}_1 - \mathbf{m}_2 \rangle \right|^2 + \int_0^{1-\varepsilon} \sup_{u \in \mathbb{S}^{d-1}} \left| h_{D_{\mu^*}^{\alpha(\beta)}}(u) - h_{D_{\nu^*}^{\alpha}}(u) \right|^2 \, \mathrm{d}\beta \\
&\quad - 2 \sup_{u \in \mathbb{S}^{d-1}} \left| \langle u, \mathbf{m}_1 - \mathbf{m}_2 \rangle \right| \int_0^{1-\varepsilon} \sup_{u \in \mathbb{S}^{d-1}} \left| h_{D_{\mu^*}^{\alpha(\beta)}}(u) - h_{D_{\nu^*}^{\alpha(\beta)}}(u) \right| \, \mathrm{d}\beta \\
&= ||\mathbf{m}_1 - \mathbf{m}_2||^2 + DR_{2,\varepsilon}^2(\mu^*, \nu^*) - 2||\mathbf{m}_1 - \mathbf{m}_2|| DR_{1,\varepsilon}(\mu^*, \nu^*).
\end{aligned}
\tag{8}
$$

Combining (7) and (8) lead to the desired result.

**Second assertion.** For any $u \in \mathbb{S}^{d-1}$, the $(1 - \alpha(\beta))$ quantiles of random variables $\langle u, X \rangle$ and $\langle u, Y \rangle$ such that $\langle u, X \rangle \sim \mathcal{N}(\langle u, \mathbf{m}_1 \rangle, u^\top \mathbf{\Sigma}_1 u)$ and $\langle u, Y \rangle \sim \mathcal{N}(\langle u, \mathbf{m}_2 \rangle, u^\top \mathbf{\Sigma}_2 u)$ are defined by

$$q_{\mu,u}^{1-\alpha(\beta)} = \langle u, \mathbf{m}_1 \rangle + \Phi^{-1}(1 - \alpha(\beta)) \sqrt{u^\top \mathbf{\Sigma}_1 u} \qquad q_{\nu,u}^{1-\alpha(\beta)} = \langle u, \mathbf{m}_2 \rangle + \Phi^{-1}(1 - \alpha(\beta)) \sqrt{u^\top \mathbf{\Sigma}_2 u},$$

where $\Phi$ is the cumulative distribution function of the univariate standard Gaussian distribution. Now, to apply Lemma A.2, it is sufficient to prove that directional quantiles are sublinear. It holds using subadditivity of the square root function. Indeed, for any $u, v \in \mathbb{S}^{d-1}$ and $\lambda > 0$, we have:

$$
\begin{aligned}
\langle u + \lambda v, \mathbf{m}_1 \rangle + \Phi^{-1}(1 - \alpha(\beta)) \sqrt{(u + \lambda v)^\top \mathbf{\Sigma}_1 (u + \lambda v)} &= \langle u, \mathbf{m}_1 \rangle + \lambda \langle v, \mathbf{m}_1 \rangle + \Phi^{-1}(1 - \alpha(\beta)) \sqrt{(u + \lambda v)^\top \mathbf{\Sigma}_1 (u + \lambda v)} \\
&\leq \langle u, \mathbf{m}_1 \rangle + \lambda \langle v, \mathbf{m}_1 \rangle + \Phi^{-1}(1 - \alpha(\beta)) \left[ \sqrt{u^\top \mathbf{\Sigma}_1 u} + \lambda \sqrt{v^\top \mathbf{\Sigma}_1 v} \right] \\
&= q_{\mu,u}^{1-\alpha(\beta)} + \lambda \, q_{\mu,v}^{1-\alpha(\beta)}.
\end{aligned}
$$

The same reasoning holds for $\nu$. Applying Lemma A.1 and Lemma A.2, for any $u \in \mathbb{S}^{d-1}$, we have $h_{D_\mu^{\alpha(\beta)}}(u) = q_{\mu,u}^{1-\alpha(\beta)}$ and $h_{D_\nu^{\alpha(\beta)}}(u) = q_{\nu,u}^{1-\alpha(\beta)}$. It follows:

$$
\begin{aligned}
DR_{1,\varepsilon}(\mu, \nu) &= \int_0^{1-\varepsilon} d_{\mathcal{H}}\left( D_\mu^{\alpha(\beta)}, D_\nu^{\alpha(\beta)} \right) \, \mathrm{d}\beta = \int_0^{1-\varepsilon} \sup_{u \in \mathbb{S}^{d-1}} \left| h_{D_\mu^{\alpha(\beta)}}(u) - h_{D_\nu^{\alpha(\beta)}}(u) \right| \, \mathrm{d}\beta \\
&= \int_0^{1-\varepsilon} \sup_{u \in \mathbb{S}^{d-1}} \left| \langle u, \mathbf{m}_1 - \mathbf{m}_2 \rangle + \Phi^{-1}(1 - \alpha(\beta)) \left[ \sqrt{u^\top \mathbf{\Sigma}_1 u} - \sqrt{u^\top \mathbf{\Sigma}_2 u} \right] \right| \, d\beta \\
&\leq ||\mathbf{m}_1 - \mathbf{m}_2|| + \int_0^{1-\varepsilon} \sup_{u \in \mathbb{S}^{d-1}} \left| \Phi^{-1}(1 - \alpha(\beta)) \left[ \sqrt{u^\top \mathbf{\Sigma}_1 u} - \sqrt{u^\top \mathbf{\Sigma}_2 u} \right] \right| \, \mathrm{d}\beta \\
&= ||\mathbf{m}_1 - \mathbf{m}_2|| + C_\varepsilon \sup_{u \in \mathbb{S}^{d-1}} \left| \sqrt{u^\top \mathbf{\Sigma}_1 u} - \sqrt{u^\top \mathbf{\Sigma}_2 u} \right|,
\end{aligned}
$$

with $C_\varepsilon = \int_0^{1-\varepsilon} \left|\Phi^{-1}(1-\alpha(\beta))\right| \, d\beta$. The lower bound is obtained by means the same reasoning. Notice that

$$\|\mathbf{m}_1 - \mathbf{m}_2\| = \sup_{u \in \mathbb{S}^{d-1}} \left|\langle u, \mathbf{m}_1 - \mathbf{m}_2\rangle\right| = \int_0^{1-\varepsilon} \sup_{u \in \mathbb{S}^{d-1}} \left|\langle u, \mathbf{m}_1 - \mathbf{m}_2\rangle\right| \, d\beta.$$

Introducing $h_{D_\mu^{\alpha(\beta)}}(u), h_{D_\nu^{\alpha(\beta)}}(u)$ and using triangular inequality, subadditivity of the supremum and linearity of the integral, we obtain:

$$\|\mathbf{m}_1 - \mathbf{m}_2\| \leq DR_{1,\varepsilon}(\mu,\nu) + C_\varepsilon \sup_{u \in \mathbb{S}^{d-1}} \left|\sqrt{u^\top \mathbf{\Sigma}_1 u} - \sqrt{u^\top \mathbf{\Sigma}_2 u}\right|,$$

which ends the proof.

### B.4 Proof of Proposition 3.7

For $DR_{p,\varepsilon}$ to break down at $\mathcal{S}_n$, it needs to have at least one trimmed-region that breaks down. Then the breakdown point of $DR_{p,\varepsilon}$ is higher than the minimum of the breakdown point of each region. Indeed, we have

$$\begin{aligned}
BP(DR_{p,\varepsilon}, \mathcal{S}_n) &= \min\left\{\frac{o}{n+o} : \sup_{Z_1,\dots,Z_o} DR_{p,\varepsilon}(\hat{\mu}_{n+o}, \hat{\mu}_n) = +\infty\right\} \\
&\geq \min_{\beta \in [0,1-\varepsilon]} \min\left\{\frac{o}{n+o} : \sup_{Z_1,\dots,Z_o} d_{\mathcal{H}}\left(D_{\hat{\mu}_{n+o}}^{\alpha(\beta,\hat{\mu}_{n+o})}, D_{\hat{\mu}_n}^{\alpha(\beta,\hat{\mu}_n)}\right) = +\infty\right\} \\
&= \min_{\beta \in [0,1-\varepsilon]} BP(D_{\hat{\mu}_n}^{\alpha(\beta,\hat{\mu}_n)}, \mathcal{S}_n).
\end{aligned}$$

Now applying Lemma 3.1 in Donoho & Gasko (1992) and Theorem 4 in Nagy & Dvořák (2021), a lower bound of the breakdown point of each halfspace region, for every $\beta \in [0, 1-\varepsilon]$, is given by

$$BP(D_{\hat{\mu}_n}^{\alpha(\beta,\hat{\mu}_n)}, \mathcal{S}_n) \geq \begin{cases} \dfrac{\lceil n\alpha(1-\varepsilon,\hat{\mu}_n)/(1-\alpha(1-\varepsilon,\hat{\mu}_n))\rceil}{n + \lceil n\alpha(1-\varepsilon,\hat{\mu}_n)/(1-\alpha(1-\varepsilon,\hat{\mu}_n))\rceil} & \text{if } \alpha(1-\varepsilon,\hat{\mu}_n) \leq \frac{\alpha_{\max}(\hat{\mu}_n)}{1+\alpha_{\max}(\hat{\mu}_n)}, \\[2em] \dfrac{\alpha_{\max}(\hat{\mu}_n)}{1+\alpha_{\max}(\hat{\mu}_n)} & \text{otherwise,} \end{cases}$$

where $\alpha_{\max}(\hat{\mu}_n) = \max_{x \in \mathbb{R}^d} HD_{\hat{\mu}_n}(x)$.

## C Approximation Algorithm

In this part, we display the approximation algorithms of the halfspace depth (see Algorithm 2), the projection depth (see Algorithm 3) and the AI-IRW depth (see Algorithm 4, proposed in Staerman et al., 2021b) used in the first step of the Algorithm 1.

---

**Algorithm 2** Approximation of the halfspace depth

---

*Initialization:* $\mathbf{X} \in \mathbb{R}^{n \times d}$, $K$.

1: Construct $\mathbf{U} \in \mathbb{R}^{d \times K}$ by sampling uniformly $K$ vectors $U_1, \dots, U_K$ in $\mathbb{S}^{d-1}$
2: Compute $\mathbf{M} = \mathbf{XU}$
3: Compute the rank value $\sigma(i,k)$, the rank of index $i$ in $\mathbf{M}_{:,k}$ for every $i \leq n$ and $k \leq K$
4: Set $D_i = \min_{k \leq K} \sigma(i,k)$ for every $i \leq n$
   **Output**: $D, \mathbf{M}$

---

---

**Algorithm 3** Approximation of the projection depth

---

*Initialization:* $\mathbf{X} \in \mathbb{R}^{n \times d}$, $K$.

1: Construct $\mathbf{U} \in \mathbb{R}^{d \times K}$ by sampling uniformly $K$ vectors $U_1, \ldots, U_K$ in $\mathbb{S}^{d-1}$
2: Compute $\mathbf{M} = \mathbf{XU}$
3: Find $\mathbf{M}_{\mathrm{med},k}$ the median value of $\mathbf{M}_{:,k}$, $\forall\, k \leq K$
4: Compute $\mathrm{MAD}_k = \mathrm{median}\{|\mathbf{M}_{i,k} - \mathbf{M}_{\mathrm{med},k}|,\ i \leq n\}$ for $k \leq K$
5: Compute $\mathbf{V}$ s.t. $\mathbf{V}_{i,k} = |\mathbf{M}_{i,k} - \mathbf{M}_{\mathrm{med},k}|/\mathrm{MAD}_k$
6: Set $D_i = \min_{k \leq K}\ 1/(1 + \mathbf{V}_{i,k})$ for every $i \leq n$
   **Output**: $D, \mathbf{M}$

---

---

**Algorithm 4** Approximation of the AI-IRW depth

---

*Initialization:* $\mathbf{X} \in \mathbb{R}^{n \times d}$, $K$.

1: Construct $\mathbf{U} \in \mathbb{R}^{d \times K}$ by sampling uniformly $K$ vectors $U_1, \ldots, U_K$ in $\mathbb{S}^{d-1}$
2: Compute $\widehat{\Sigma}$ using any estimator
3: Perform Cholesky or SVD on $\widehat{\Sigma}$ to obtain $\widehat{\Sigma}^{-1/2}$
4: Compute $\mathbf{V} = \widehat{\Sigma}^{-1/2}\mathbf{U}/||\widehat{\Sigma}^{-1/2}\mathbf{U}||$
5: Compute $\mathbf{M} = \mathbf{XV}$
6: Compute the rank value $\sigma(i,k)$, the rank of index $i$ in $\mathbf{M}_{:,k}$ for every $i \leq n$ and $k \leq K$
7: Set $D_i = \frac{1}{K}\sum_{k=1}^{K}\ \sigma(i,k)$ for every $i \leq n$
   **Output**: $D, \mathbf{M}$

---

# D  Additional Experiments

## D.1  Illustration of Data Depth Contours

Figure 6, which plots a family of AI-IRW (using MCD estimator) depth induced trimmed-contours for a dataset contaminated with outliers, illustrates its robustness.

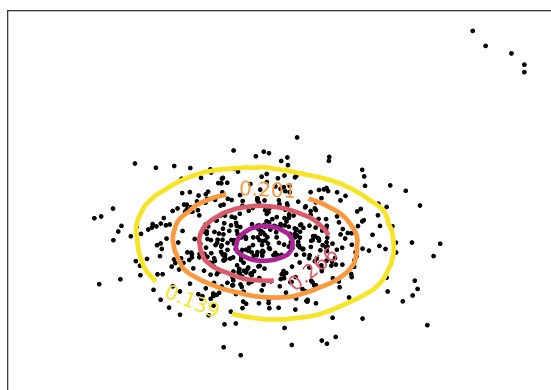

Figure 6: AI-IRW depth contours for a bivariate sample contaminated with outliers.

## D.2  Illustration of the Depth Trimmed-Regions based Pseudo-Metric

Figure 1, which plots a family of (approximated) AI-IRW depth induced trimmed-regions for two datasets contaminated with outliers, illustrates the key idea of our proposed pseudo-metric.

### D.3 Empirical Analysis of Statistical Rates

This part presents complementary results of those obtained in the Section 5.1. Considering the same experiment as in the core paper, Figures 7 and 8 display the results of the same experiment but with dimension $d = 5$ and $d = 10$, respectively.

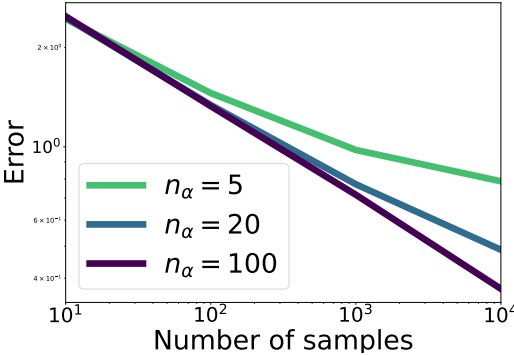
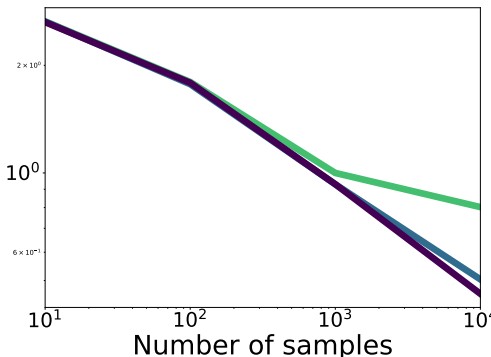

Figure 7: Empirical analysis of statistical convergence rates. Resulting error of the proposed pseudo-metric when increasing the sample size using the projection depth (left) and the halfspace depth (right) for various $n_\alpha$ parameters with $d = 5$.

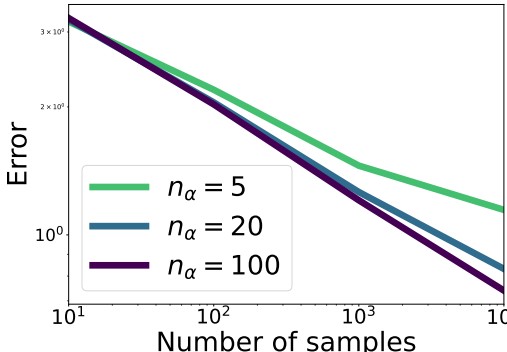
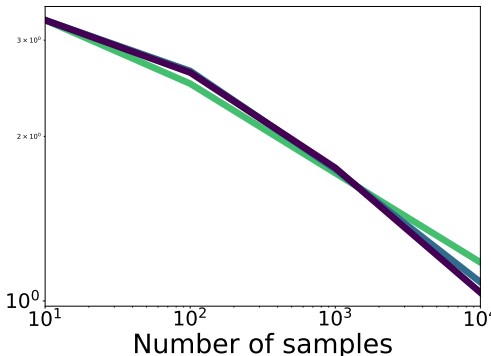

Figure 8: Empirical analysis of statistical convergence rates. Resulting error of the proposed pseudo-metric when increasing the sample size using the projection depth (left) and the halfspace depth (right) for various $n_\alpha$ parameters with $d = 10$.

### D.4 The Influence of the Parameter $\varepsilon$

The parameter $\varepsilon$ plays the role of the robust tuning parameter of $DR_{2,\varepsilon}$. In this part, we complete our theoretical results provided in Section 3.2. We assess the robustness of our pseudo-metric making varying the parameter $\varepsilon$. Precisely, we simulate two normal samples $\mathbf{X}$ and $\mathbf{Y}$ from two standard Gaussian distributions in dimension two with a sample size of 10000. From that, we construct abnormal samples with a proportion of anomalies equal to $\{1\%, 10\%, 20\%\}$. To that end, we choose a proportion of normal samples and replace their first (for $\mathbf{X}$) and second (for $\mathbf{Y}$) coordinates as follows: $X_{\mathrm{anom}} = 30 + 50Z$ and $Y_{\mathrm{anom}} = -30 - 50Z$ where $Z$ follows a uniform distribution on $[0,1]$; leading to points far from the normal distributions. Thus, we compute $DR_{2,\varepsilon}$ with both robust and non-robust data depths, i.e. the projection and halfspace depths between $\mathbf{X}$ and $\mathbf{Y}$ being used as a benchmark. Further, we compute $DR_{2,\varepsilon}$ between abnormal samples and report mean error (comparing values obtained between normal samples and values obtained between

abnormal samples; averaged over ten runs) on Figure 9. First, when computing with a robust depth function, we can see that the robustness of the proposed pseudo-metric relies directly on the parameter $\varepsilon$. This is shown by the presence of an elbow when the parameter $\varepsilon$ reaches the level of the proportion of anomalies. In contrast, we can see that for a non-robust depth function such as the halfspace depth, our proposed pseudo-metric becomes non-robust once the abnormal proportion is higher than 1%, leading to a poorly robust depth. This experiment then confirms our theoretical results on the Breakdown Point of $DR_{p,\varepsilon}$ highlighted in Propostion 3.7. The parameter $\varepsilon$ provides robustness to our pseudo-metric when combined with a robust depth function.

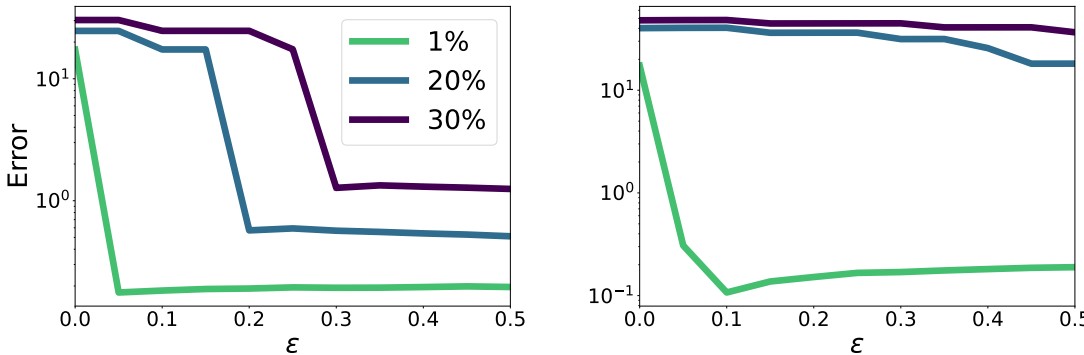

Figure 9: Influence of the parameter $\varepsilon$ on the robustness of the proposed pseudo-metric with a robust depth function (the projection depth, left) and a non-robust one (the halfspace depth, right) for various proportion of anomalies.

### D.5 The Choice of the Parameter $n_\alpha$

Proposition 3.6 allows to derive a closed form expression for $DR_{2,\varepsilon}(\mu,\nu)$ when $\mu,\nu$ are Gaussian distributions with the same variance-covariance matrix. In order to investigate the quality of the approximation on light-tailed and heavy-tailed distributions, we focus on computing $DR_{2,0.1}$ (with $K = 500$) for varying number of $n_\alpha$ between a sample of 1000 points stemming from $\mu \sim \mathcal{N}(\mathbf{0}_d, \Sigma)$ for $d \in \{2, 3, 10\}$, $\Sigma$ drawn from the Wishart distribution (with parameters $(d, I_d)$) on the space of definite matrices and three different samples (which yields nine settings). These three samples are constructed from 1000 observations stemming from elliptically symmetric *Cauchy*, *Student-$t_2$* and *Gaussian* distributions all centered at $\mathbf{7}_d$. Results that report the averaged approximation error and the 25-75% empirical quantile intervals are depicted in Figure 10. They show that $DR_{p,\varepsilon}$ converges slowly for *Cauchy* with growing $n_\alpha$, while it converges with small $n_\alpha$ for *Gaussian* and *Student-$t_2$* distributions.

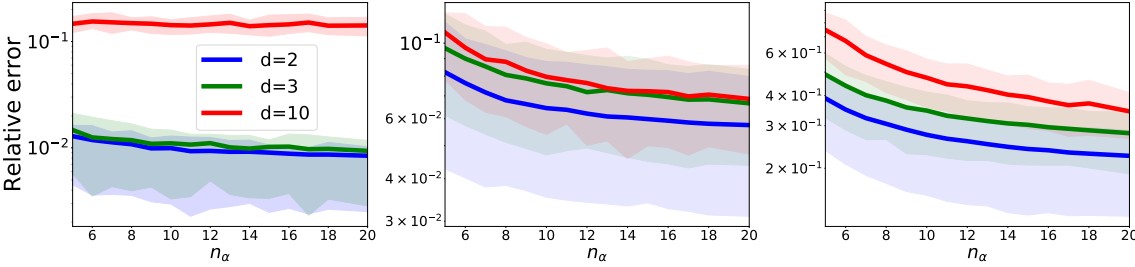

Figure 10: Relative approximation error (averaged over 100 repetitions, y-axis in log scale) of $DR_{p,\varepsilon}$ for elliptically symmetric *Cauchy* (left), *Student-$t_2$* (middle) and *Gaussian* (right) distributions for differing numbers of $n_\alpha$.

### D.6 Robustness to Outliers

Datasets on which experiments regarding "Robustness to outlier" in Section 5 have been performed are displayed in Figure 11.

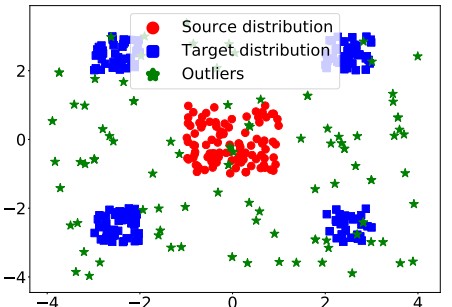 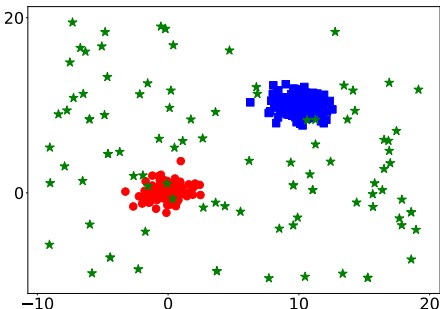

Figure 11: datasets related to robustness experiments depicted in Section 5 with 20% of outliers for *fragmented hypercube* (left) and *Gaussian* (right).

### D.7 Trimming approximation comparison between $DR_{p,\varepsilon}$ and $TSW_{p,\varepsilon}$

This part investigates the effect of the trimming parameter for $DR_{p,\varepsilon}$ and the trimmed sliced Wasserstein, $TSW_{p,\varepsilon}$, regarding the stability and efficiency of the trimming. We simulate two datasets of size $n = 100$ stemming from a $d$-dimensional centred Gaussian distribution with $d \in \{3, 10\}$ and $\Sigma$ drawn from a Wishart distribution. First, We compute the two metrics with $p = 2$ and $\varepsilon = 0$. Further, we compute $DR_{2,\varepsilon}$ and $TSW_{2,\varepsilon}$ for $\varepsilon \in [0.05, 0.45]$ and compute the absolute differences with the non-trimmed values, i.e. $|DR_{2,0} - DR_{2,\varepsilon}|$ and $|TSW_{2,0} - TSW_{2,\varepsilon}|$. The experiment is repeated 10 times, and the results are reported in Figure 12. While the trimming effect does not depend on the dimension of the trimmed sliced Wasserstein, the trimming effect drastically deteriorates the given value of the sliced Wasserstein. In contrast, the trimming effect on $DR$ deteriorates much less the metric quality, even if this increases with the dimension.

### D.8 Illustration different depths and outliers

This part provides intuition between three data depth, IRW, AI-IRW and the robust version of AI-IRW regarding how they discard abnormal data. The experiment is conducted as follows. We simulate a two dimensional Gaussian distribution and add some *isolated* anomalies (orange) and *aggregated* anomalies (red) at hand. We compute the three depth functions on this dataset and draw the several quantile regions, see Figure 13. Quantiles regions defined by each depth are of different shape, characterizing abnormal data in different manners. Regarding the pseudo-metric $DR_{p,\varepsilon}$, discarding abnormal data will depends on (1) the quantiles regions defined by the chosen data depth and (2) the trimming parameter $\varepsilon$. The quality of the $DR_{p,\varepsilon}$ in discarding anomalies relies on the quality of the chosen depth functions that are gathererd, e.g. in Mosler & Mozharovskyi (2022).

## E Application to NLP

In this section, we gather details on experimental settings and additional results on the automatic evaluation of natural language generation (NLG).

### E.1 Extended related works on Automatic Evaluation of NLG

Many metrics have been recently introduced for the automatic evaluation of text generation. In this work, we rely on untrained metrics. These metrics can be grouped into two categories: string-based metrics that

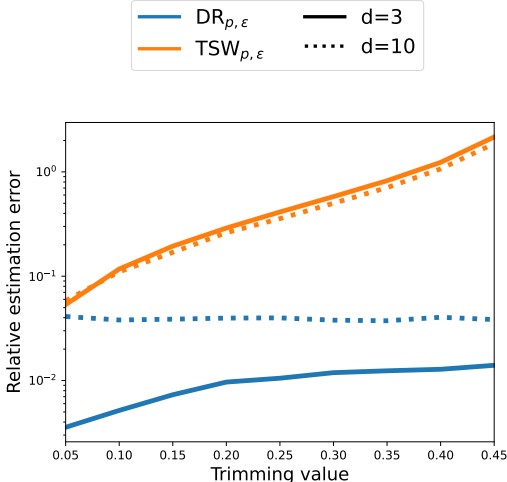

Figure 12: Approximation error between the metric computed with $\varepsilon = 0$ (without trimming) and $\varepsilon$ varying in $[0.05, 0.45]$ for both $DR_{p,\varepsilon}$ and $TSW_{p,\varepsilon}$.

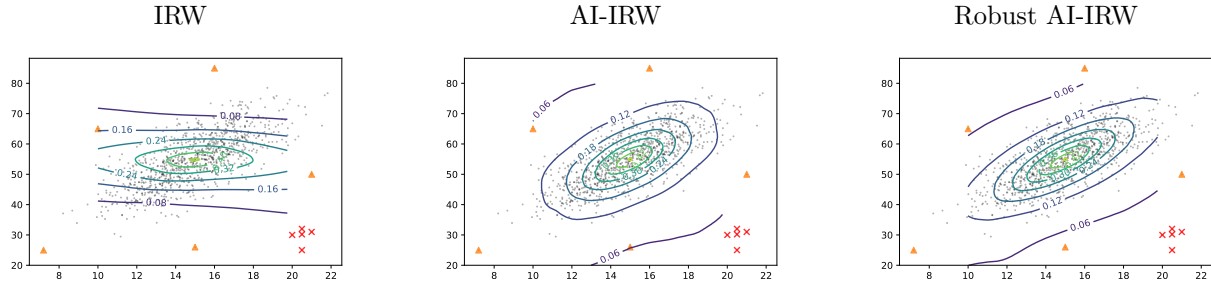

Figure 13: Contours of different depths computed on data with outliers, IRW (left), AI-IRW (middle) and the robust version of AI-IRW (right). Red crosses and orange triangles are two different type of outliers.

depend on the string representation of the input texts to compute the similarity score and embedding-based metrics that rely on a continuous representation of the texts.

String matching metrics can be divided into two categories: N-gram matching and edit distance-based metrics. Perhaps the most used N-gram matching metrics are BLEU, ROUGE and METEOR. Edit distance-based metrics (e.g. TER; Snover et al., 2006) measure the distance as the number of basic operations such as 'edit'/'delete'/'insert'. Variants of TER include CHARACTERE (Wang et al., 2016), CDER (Leusch et al., 2006), EED (Stanchev et al., 2019). String-based metrics fail to produce meaningful scores in the case of paraphrases, especially if no common n-grams are found between the candidate and the reference text.

The second category of untrained metrics (namely embedding-based metrics) achieves state-of-the-art performance in many NLG evaluation tasks and has been introduced to address the issues mentioned above. Originally introduced for the widely used words embedding (Garcia et al., 2019; Colombo et al., 2019; 2020; 2021b) such as Word2Vec (Mikolov et al., 2013) or Glove (Pennington et al., 2014), this class of metrics has leveraged recently introduced contextualized word representations (CWR). CWR such as BERT, ELMO (Peters et al., 2018), HILAMOD (Chapuis et al., 2020; 2021) or ROBERTA (Liu et al., 2019b) are popular in NLP (Witon et al., 2018) as they achieve SOTA performance on many tasks. The two most popular metrics are MoverScore and BertScore.

### E.2 Evaluation

For the task of evaluation of text generation, we assume that we have access to a dataset $\{T_{R_i}, \{T_{G_i}^j, h(T_{G_i}^j)\}_{j=1}^{n_S}\}_{i=1}^{n_T}$ where $T_{G_i}^j$ represents the $i$-th generated text by the $j$-th natural generation system, and $h(T_{G_i}^j)$ represents score assigned by the human annotator[1] to $T_{G_i}^j$, and $T_{R_i}$ is the reference text. $n_T$ is the number of available texts, and $n_S$ is the number of different systems.

To assess the relevance of an evaluation metric $\mathfrak{M}$, the correlation with the human judgment is considered one of the most important criteria (Banerjee & Lavie, 2005; Koehn, 2009; Chatzikoumi, 2020). To measure this correlation, two evaluation strategies are commonly adopted and built on top of a classical correlation measure, denoted $C$, e.g. Kendall ($\tau$; Kendall, 1938), Pearson ($r$; Leusch et al., 2003) or Spearman ($\rho$; Melamed et al., 2003).

- *The text level correlation* ($C_{text}$) measures the ability of the metric to distinguish between badly and well generated text. Formally, $C_{text}$ is defined as follows:

$$C_{text} = \frac{1}{N_T} \sum_{i=1}^{n_T} C(\mathbf{M}_i^{text}, \mathbf{H}_i^{text}),$$  (9)

$$\mathbf{M}_i^{text} = \left[\mathfrak{M}(T_{R_i}, T_{C_i}^1), \cdots, \mathfrak{M}(T_{R_i}, T_{C_i}^{n_S})\right],$$

$$\mathbf{H}_i^{text} = \left[h(T_{C_i}^1), \cdots, h(T_{C_i}^{n_S})\right].$$

- *The system level correlation* ($C_{sys}$) assesses the ability of a metric to distinguish between good and bad systems. Formally, $C_{sys}$ is defined as follows:

$$C_{sys} = C(\mathbf{M}^{sys}, \mathbf{H}^{sys}),$$  (10)

$$\mathbf{M}^{sys} = \left[\frac{1}{n_T} \sum_{i=1}^{n_T} \mathfrak{M}(T_{R_i}, T_{C_i}^1), \cdots, \frac{1}{n_T} \sum_{i=1}^{n_T} \mathfrak{M}(T_{R_i}, T_{C_i}^{n_S})\right],$$

$$\mathbf{H}^{sys} = \left[\frac{1}{n_T} \sum_{i=1}^{n_T} h(T_{C_i}^1), \cdots, \frac{1}{n_T} \sum_{i=1}^{n_T} h(T_{C_i}^{n_S})\right],$$

We refer the reader to Bhandari et al. (2020) for further details on the evaluation of text generation.

---

[1] In practice an averaged score is considered as each sentence is annotated by 3 different annotators. The considered datasets directly provide the aggregated score.

| | Correctness | | | Data Coverage | | | Relevance | | |
|---|---|---|---|---|---|---|---|---|---|
| | $r$ | $\tau$ | $\rho$ | $r$ | $\tau$ | $\rho$ | $r$ | $\tau$ | $\rho$ |
| $DR_{p,\varepsilon}$ | **89.4** | **80.0** | **92.6** | **84.2** | 58.3 | 72.3 | **86.2** | 62.7 | 72.9 |
| Wasserstein | 86.2 | 73.0 | 86.7 | 80.4 | 45.3 | 62.3 | 83.8 | 51.3 | 67.6 |
| Sliced-Wasserstein | 86.1 | 73.0 | 85.8 | 80.9 | 45.5 | 60.0 | 82.0 | 51.3 | 68.2 |
| MMD | 25.4 | 71.7 | 8.3 | 19.1 | 45.3 | 10.0 | 26.1 | 51.3 | 15.0 |
| BertScore | 85.5 | 73.3 | 83.4 | 74.7 | 53.3 | 68.2 | 83.3 | 65.0 | **79.4** |
| MoverScore | 84.1 | 73.3 | 84.1 | 78.7 | 53.3 | 66.2 | 82.1 | 65.0 | 77.4 |
| BLEU | 77.6 | 60.0 | 66.3 | 55.7 | 36.6 | 50.2 | 63.0 | 51.6 | 65.2 |
| ROUGE-1 | 80.6 | 65.0 | 65.0 | 76.5 | **60.3** | **76.3** | 64.3 | 56.7 | 69.2 |
| ROUGE-2 | 73.6 | 58.3 | 63.3 | 54.7 | 35.0 | 43.1 | 62.0 | 46.7 | 60.8 |
| METEOR | 86.5 | 70.0 | 66.3 | 77.3 | 46.6 | 50.2 | 82.1 | 58.6 | 65.2 |
| TER | 79.6 | 58.0 | 78.3 | 69.7 | 38.0 | 58.2 | 75.0 | **77.6** | 70.2 |

Table 4: WebNLG 2020 (full results): absolute correlation at the system level with three human judgment criteria. Best overall results are indicated in bold, best results in their group are underlined.

### E.3 Results on Data2text

In this section, we gather further details and results on data2text automatic evaluation.

### E.3.1 Task Description

In WebNLG 2020, the goal is to create new efficient generation algorithms that can verbalise knowledge-based fragments. These algorithms are called Knowledge Base Verbalizers (Gardent et al., 2017) and are used during the micro-planning phase of NLG systems (Ferreira et al., 2018). WebNLG has been gathered to be more representative of the progress of recent NLG systems than previously existing task-oriented dialogue datasets (see e.g. SFHOTEL (Wen et al., 2015) and BAGEL (Mairesse et al., 2010)). As previously mentioned for the data2text task we work on the WebNLG2020 challenge (Gardent et al., 2017; Perez-Beltrachini et al., 2016). Data and system performances can be found in `https://webnlg-challenge.loria.fr/`. The task consists in mapping RDF triples to natural language (RDF format is used for many application including FOAF (see `http://www.foaf-project.org/`). For WebNLG 2020, the triplets are extracted from DBpedia (Auer et al., 2007). Data have been made freely available from the authors at `https://gitlab.com/shimorina/webnlg-dataset/-/tree/master/release_v3.0`. To compose this dataset, 15 systems (both symbolic and neural-based) have been used. The final dataset is composed of over 3k samples of human annotations (see `https://webnlg-challenge.loria.fr/files/WebNLG-2020-Presentation.pdf` for more details).

**Example**: Given the following triplet `(John_Blaha birthDate 1942_08_26) (John_Blaha birthPlace San_Antonio) (John_Blaha job Pilot)` the ground-truth reference is `John Blaha, born in San Antonio on 1942-08-26, worked as a pilot.`

### E.3.2 Results

We gather in Table 4 complete results on the WebNLG tasks including results on ROUGE-2. To compare $DR_{p,\varepsilon}$ (with $\varepsilon = 0.01$, $n_\alpha = 5$, $p = 2$) with the different metrics (i.e. Wasserstein, Sliced-Wasserstein, MMD), we work on Roberta-based model from the HuggingFace hub (Wolf et al., 2019) and extract representation from the 11th layer. From Table 4, we observe a similar behavior from BertScore and MoverScore. This similarity has also been reported in a different setting in the previous work of Zhao et al. (2019). Overall, we observe that $DR_{p,\varepsilon}$ is always among its group's top-scoring metrics and achieves the best overall results on several configurations. It is worth noticing that $DR_{p,\varepsilon}$ only relies on information available in the candidate and the reference text. In contrast, BertScore and MoverScore use IDF information computed on every dataset.

### E.4 Results on Summarization

In this section, we gather experimental details and results on the automatic evaluation of the text summarization task.

#### E.4.1 Task Description

Text summarization has attracted much attention in recent years (Zhang et al., 2020). Two types of models exist: *extractive* and *abstractive*. In extractive summarization, the system copies chunks of informative fragments from the input texts, whereas, in abstractive summarization, the system generates novel words. In this section, we describe our experimental setting. We present the tasks and the baseline metrics used for the automatic evaluation of summarization. We work with the dataset from Bhandari et al. (2020) for this task. This dataset has been introduced to solve several flaws (Rankel et al., 2013) present in existing summarization datasets such as TAC (Dang & Owczarzak, 2008; McNamee & Dang, 2009). The dataset has been annotated using the pyramid score (Nenkova et al., 2007; Nenkova & Passonneau, 2004) and automatically built from the CNN/Daily News (Bhandari et al., 2020). It gathers 11 490 summaries coming from 11 extractive systems (See et al., 2017; Chen & Bansal, 2018; Raffel et al., 2019; Gehrmann et al., 2018; Dong et al., 2019; Liu & Lapata, 2019; Lewis et al., 2019; Yoon et al., 2020) and 14 abstractive systems (Zhou et al., 2018; Narayan et al., 2018; Kedzie et al., 2018; Zhong et al., 2019; Liu & Lapata, 2019; Dong et al., 2019; Wang et al., 2020; Zhong et al., 2020).

**Example**: The goal is to assign a similarity score between a reference text: "*Manchester United take on Manchester City on Sunday. Match will begin at 4 pm local time at United's Old Trafford home. Police have no objections to kick-off being so late in the afternoon. Last late afternoon weekend kick-off in the Manchester derby saw 34 fans arrested at Wembley in 2011 fa cup semi-final*" and the text generated by a NLG system: "*Manchester Derby takes place at Old Trafford on Sunday afternoon police have no objections to the late afternoon kick-off both sides are challenging for a top-four spot in the Premier League the man in charge of patrolling the sell-out clash has no such fears*".

#### E.4.2 Results

We gather in Table 5, the results on the summarization task. We use a bert-based uncased model and rely on the representations extracted from the 9th layer (similarly to BertScore). For this experiment the following parameters are used: $\varepsilon = 0.01$, $n_\alpha = 5$, $p = 2$. For this task, we can reproduce results from Bhandari et al. (2020) where the different behavior regarding the extractive and the abstractive systems is also observed. In this experiment, we observe that $DR_{p,\varepsilon}$ can achieve stronger results than other metrics based on Wasserstein, Sliced-Wasserstein and MMD. We also observe that $DR_{p,\varepsilon}$ outperforms MoverScore and BertScore on extractive systems (on $r$ and $\tau$). We believe these results support our approach.

|  | Abstractive | | | Extractive | | |
|---|---|---|---|---|---|---|
|  | $r$ | $\tau$ | $\rho$ | $r$ | $\tau$ | $\rho$ |
| $DR_{p,\varepsilon}$ | 72.1 | 72.1 | 70.1 | 91.5 | **91.5** | 69.2 |
| Wasserstein | 71.0 | 70.4 | 71.1 | 74.2 | 74.2 | 40.0 |
| Sliced-Wasserstein | 70.1 | 68.7 | 71.0 | 72.4 | 73.9 | 69.2 |
| MMD | 68.2 | 67.5 | 67.9 | 75.6 | 75.6 | 56.1 |
| BertScore | 71.7 | 71.9 | 72.0 | 70.9 | 72.9 | **73.8** |
| MoverScore | 72.4 | 71.9 | 73.0 | 76.1 | 76.1 | 47.4 |
| ROUGE-1 | **73.5** | 73.0 | **74.4** | 72.2 | 74.0 | 69.1 |
| ROUGE-2 | 73.0 | **73.5** | 73.0 | 55.1 | 53.2 | 69.1 |
| JS-2 | 68.9 | 6.8 | 69.8 | **92.9** | 5.5 | 19.0 |

Table 5: Summarization: absolute correlation coefficients (using Pearson ($r$), Spearman ($\rho$) and Kendall ($\tau$) coefficient) between different metrics on text summarization. Best overall results are indicated in bold, best results in their group are underlined.

