# OpenReview forum: "A Pseudo-Metric between Probability Distributions based on Depth-Trimmed Regions"
_TMLR — Accepted by TMLR_

### Review · Reviewer_vUsG · 2023-11-06

**Summary Of Contributions:**

The paper develops a new pseudo-metric between probability distributions based on the notion of 'data depth', which provides an ordered series of subsets of the input space. The empirical behavior of the metric is studied in small case studies, and two demonstration applications are proposed.

**Audience:**

Yes

**Broader Impact Concerns:**

The paper is on the abstract-and-conceptual side of the ML research spectrum, so I have no concerns in this regard.

**Claims And Evidence:**

Yes

**Requested Changes:**

I would like for manuscript to be sufficiently clear that my above questions can be answered.

**Strengths And Weaknesses:**

## Strengths
* The paper investigates a broadly applicable problem, namely computing similarities between distributions. As such, the paper should have a readership as required by the TMLR policy.
* The experiments highlight the empirical statistical robustness of the proposed similarity measure.

## Weakness
* As a non-expert, I find that the main issue with the paper is that the paper is rather difficult to follow. I detail some of these issues below.
* The paper could be much improved by including attempts to visualize the key ideas.
* Equation 1 is very generic, which implies that the subsequent discussion is quite difficult for readers who are not already familiar with data depths (I am not).
* It is difficult to determine what I am supposed to learn from the studied machine learning 'applications'. The first (clustering) seems to be very much a toy-solution, which is not representative of how one would solve such a problem. The second (NLP) is beyond my area of expertise. My key concern is that I do not know what I am supposed to learn from these experiments (what is the conclusion?)


## Questions
(these are listed in the order my questions appeared while reading the paper)
* Page 1 (minor): what is an 'IPM'?
* Page 3: What should I learn from the second line of Eq 1? Isn't it trivial that a function is a mapping?
* Page 3: How do I derive from Eq 1 that "data depth naturally and in a nonparametric way defines a pre-order on $\mathbb{R}^d$"? I do not doubt that the statement is true, but does it perhaps require more than Eq. 1?
* Page 3 and 4: you consider convex depth functions, but I find it very unclear, which practical limitations this places on the associated distribution. I mean, when would I ever want a convex depth function alongside a non-convex probability density function? Eq. 2 seems to suggest that you indeed consider convex PDFs.
* Page 3 (minor): why are the postulates "D1 -- D4" named "D<number>"? What is the "D" short for?
* Page 4: "To the best of our knowledge, the supremum exists..." Does this statement imply that you do not have proof of the statement and have been unable to locate one in the literature?
* Page 6: "This proposition shows that $DR_{2, \epsilon}$ is able to factor..." So the proposition does not provide a guarantee? In this case, why is the proposition even interesting?
* Page 10 (minor): What does "$\mathbf{10}_2$" mean?

---

> ### Author Response · Authors · 2024-01-21
> **Answer to Reviewer vUsG**
>
> We thank the reviewer for their careful review and their constructive comments.
>
> “Page 1 (minor): what is an 'IPM'?"
>
> * Integral Probability Metrics, it has been clarified in the new version.
>
> “Page 3: What should I learn from the second line of Eq 1? Isn't it trivial that a function is a mapping?"
>
> * Equation (1) is indeed a generic formulation for data depth. However, as detailed in Section 2, a function from this equation has to satisfy properties of D1-D4 to be considered a convex depth, drastically limiting the function possibilities.
>
> “Page 3: How do I derive from Eq 1 that "data depth naturally and in a nonparametric way defines a pre-order on "? I do not doubt that the statement is true, but does it perhaps require more than Eq. 1?"
>
> * Indeed, the Equation (1) alone does not necessarily prove this. However, the condition D1-D4 does define a pre-order. Considering the nonparametric way, it is the way they all have been defined in the literature, but there is no limitation in designing parametric ones.
>
> “Page 3 and 4: you consider convex depth functions, but I find it very unclear, which practical limitations this places on the associated distribution. I mean, when would I ever want a convex depth function alongside a non-convex probability density function? Eq. 2 seems to suggest that you indeed consider convex PDFs. "
>
> * This pseudo-metric may be defined for non-convex depth functions but may not be computable in practice. Indeed, the computationally efficient approximation proposed in Section 4 is valid only for convex depth.
>
> “Page 3 (minor): why are the postulates "D1 -- D4" named "D<number>"? What is the "D" short for?"
>
> * D is for depth. We could have named them P1–P4 but it would be confusing with the probability also denoted by P.
>
> “Page 4: "To the best of our knowledge, the supremum exists..." Does this statement imply that you do not have proof of the statement and have been unable to locate one in the literature?"
>
> * It means that the existence of the supremum has not been theoretically demonstrated for any depth function and any probability distribution even if it always exists with empirical distribution. We provide references for those that have been proved so far.
>
> “Page 6: "This proposition shows that it is able to factor..." So the proposition does not provide a guarantee? In this case, why is the proposition even interesting?"
>
> * It means ‘’that it can factor…’’. The proposition 3.6 is true for any probability distribution that satisfy the assumption of the proposition.
>
> “Page 10 (minor): What does "" mean?"
>
> * $\mathbf{10}_2$ is the vector [10, 10]. It has been clarified in the experiment of the latest version.

---

> > ### Comment · Reviewer_vUsG · 2024-01-23
> > **Thanks for the clarifications**
> >
> > I appreciate the replies. Thanks.

---

### Review · Reviewer_bLUo · 2023-11-22

**Summary Of Contributions:**

The authors propose a new pseudo-metric for comparing absolutely continuous probability measures, based on the Hausdorff distance between the depth regions of two measures, extending the notion of one-dimensional quantile to the multivariate framework. Depth regions are constructed from convex data depth functions, which are characterized by their support functions, enabling to rewrite the Hausdorff distance in a more convenient way. The proposed discrepancy is designed to enforce robustness, by discarding the tail of the distributions, as a function of a hyperparameter. The usefulness of this discrepancy is evaluated in a clustering task and an automatic evaluation of language generation.

**Audience:**

Yes

**Broader Impact Concerns:**

I don't think this paper requires a Broader Impact Statement.

**Claims And Evidence:**

Yes

**Requested Changes:**

- Adding of a table (or equivalent) to summarize the properties of the discrepancy depending on the type of data depth function (or depth functions themselves), in terms of robustness, computation time of the approximation algorithm, breakpoint for example, could help clarifying the different contributions. In particular, the proposed discrepancy is not a metric in general as $DR_{p,\varepsilon}(\mu,\nu)=0$ does not imply $\mu=\nu$. However, I suppose this could be the case for particular convex data depth functions?

- I would have find interesting to analyze the discrepancy (in simple cases, e.g. elliptic distributions) depending on the convex data depth functions, to better understand which outliers are discarded for example.

- I may have missed something, but the depth-trimmed regions discrepancy measure (Def 3.1) is defined for absolutely continuous measures, whereas in the robustness result (Prop 3.7), the discrepancy is considered between empirical measures. Could you clarify this point?

In my opinion, if manageable, these three points are important to strengthen the paper.

**Strengths And Weaknesses:**

Strengths:
- The article is clear and the concepts of depth regions and data depth functions are well presented, as well as the motivation behind the proposed discrepancy. Indeed, for convex data depth functions and elliptical distributions, the depth-trimmed regions can be compared to the density level sets, allowing to better appreciate the definition of the discrepancy. In particular, the divergence, defined to be robust to outliers (a claim supported by a theoretical result), is relevant to both the statistical community and in machine learning tasks.
- The detailed bibliography on related discrepancies for absolutely continuous measures is significant. The choices of the metrics to be compared with in the experiments (which may or may not handle outliers) is also relevant in my opinion.
- An approximation method for calculating the discrepancy is proposed and shows good time complexity. The codes to reproduce the experiments in the paper are provided.


Weaknesses:
- The main weakness in my opinion is the lack of theoretical study of statistical rates of convergence when dealing with empirical probability measures (an analysis which must be quite difficult to develop, due to the Hausdorff distance). Nevertheless, an empirical analysis of statistical rates is proposed. Would it be possible to have some insights on this for particular cases (e.g. elliptical distributions and projection depth)?

---

> ### Author Response · Authors · 2024-01-21
> **Answer to Reviewer bLUo**
>
> We thank the reviewer for their constructive comments and for acknowledging our efforts to validate the approach. Regarding the concerns of the reviewer, we answered point by point below.
>
> “Adding of a table (or equivalent) to summarize the properties of the discrepancy depending on the type of data depth function (or depth functions themselves), in terms of robustness, computation time of the approximation algorithm, breakpoint for example, could help clarifying the different contributions. In particular, the proposed discrepancy is not a metric in general as $DR_{p, \varepsilon}(\mu, \nu)=0$ does not imply  $\mu=\nu$. However, I suppose this could be the case for particular convex data depth functions?"
>
> * We agree. We added a table, in Section A.4 in the Appendix, that summarizes the characteristic of $DR_{p, \varepsilon}$ according to the known properties of the depth functions the three following functions: ‘halfspace’, ‘projection’, and ‘AI-IRW’. It is worth noting that the definition of our pseudo-metric is general and may use any convex data depth. Therefore, the metric's properties directly rely on existing results on the depth functions.
> Indeed, in some cases of convex data depth, the pseudo-metric could define a distance. $DR_{p,\varepsilon}$ is a distance if and only if the upper-level sets of the chosen data depth fully characterize probability distributions. To our knowledge, It has been proved only for the Halfspace depth under mild assumptions [7, 8] and the Zonoid depth assuming the first moment on the distribution [9]. This remark has been added in Section A.4. of the new manuscript.
>
> “I would have find interesting to analyze the discrepancy (in simple cases, e.g. elliptic distributions) depending on the convex data depth functions, to better understand which outliers are discarded for example."
>
> * Due to the construction of our pseudo-metric, outliers are discarded by the quantile regions designed by the chosen depth function. To provide some intuition, we added an experiment in Section D.9 of the Appendix of the new version. It illustrates the different quantile regions each depth defines for a Gaussian distribution with additional abnormal data. Other regions are drawn depending on the chosen depth function. Thus, the pseudo-metric, relying on the Hausdorff distance of these regions, will provide different values of $DR_{p, \varepsilon}$. Considering the number of depth functions available, it is generally difficult to know precisely which depth is better in a specific situation. However, an extensive comparison has been recently made in  [6]. We refer the reader to Figures  1,2 and 4 of this paper, where quantile regions are drawn for an extensive number of depth functions, giving insight into which depth functions to choose regarding the situation.
>
> “I may have missed something, but the depth-trimmed regions discrepancy measure (Def 3.1) is defined for absolutely continuous measures, whereas in the robustness result (Prop 3.7), the discrepancy is considered between empirical measures. Could you clarify this point?"
>
>
> * We thank the reviewer for notifying us of this point. After careful verification of the proofs, the absolute continuous property is not needed, and all our results remain valid for any probability measure on $\mathbb{R}^{d}$. We then generalized our definition to any probability measure on $\mathbb{R}^{d}$ in the new version of the paper.
>
>
> **References:**
>
> [6] Mosler, K., & Mozharovskyi, P. (2022). Choosing among notions of multivariate depth statistics. Statistical Science, 37(3), 348-368.
>
> [7] Hassairi, A., & Regaieg, O. (2008). On the Tukey depth of a continuous probability distribution. Statistics & probability letters, 78(15), 2308-2313.
>
> [8] Kong, L., & Zuo, Y. (2010). Smooth depth contours characterize the underlying distribution. Journal of Multivariate Analysis, 101(9), 2222-2226.
>
> [9]  Mosler, K. (2002). Multivariate dispersion, central regions, and depth: the lift zonoid approach (Vol. 165). Springer Science & Business Media.

---

> > ### Comment · Reviewer_bLUo · 2024-01-26
> > **Response to the authors**
> >
> > I would like to thank the authors for their precise responses to my comments and to the questions raised by other reviewers. The changes made to the paper are satisfactory and I recommend the acceptance of this paper.

---

### Review · Reviewer_vcJR · 2024-01-06

**Summary Of Contributions:**

This manuscript proposes a new family of divergences over the space of probability measures, which generalize the univariate Wasserstein distance. These divergences are defined by the Hausdorff metric between the depth regions of the two distributions, integrated over all scales. In this definition, extreme scales are omitted, which makes the divergences robust to outliers.

The authors verify certain elementary metric properties of their proposed divergences, and they derive several results connecting their divergences to the Wasserstein distance. They also quantify the robustness of their divergences via the traditional notion of breakdown point. The authors devise a clever algorithm for computing their divergence, by noting that the Hausdorff distance between convex depth regions can be efficiently computed. They then conduct a large range of simulations and numerical applications.

**Audience:**

Yes

**Broader Impact Concerns:**

No broader impact concerns.

**Claims And Evidence:**

Yes

**Requested Changes:**

- As per my previous comment, I encourage the authors to expand their discussion of the possible benefits of their metric as compared to (trimmed) Wasserstein-type metrics.
 - I also encourage the authors to consider using the trimmed Wasserstein metric (or its sliced, max-sliced, etc. versions) in their numerical comparisons. I believe this would provide a more fair comparison in Figure 5.
- On page 8, the authors comment that "deriving theoretical finite-sample analysis may appear to be challenging" --- could the authors elaborate on this comment? What makes it challenging to analyze estimators of the DR metric?
- On page 9, could the authors please elaborate on their comment that the sliced-Wasserstein Monte Carlo approximation degrades exponentially in the dimension? Reference [5, Theorem 6] would suggest otherwise.

Typographical comments:
 - On page 4, replaced "bever" by "Bever".
 - On page 6, please put the condition o \in \mathbb{N}^* inside the curly braces of the minimum.
 - On page 8, n^{1/4} should be n^{-1/4}, and similarly for the subsequent rates.


[5] Nadjahi, K., Durmus, A., Chizat, L., Kolouri, S., Shahrampour, S., & Simsekli, U. (2020). Statistical and topological properties of sliced probability divergences. Advances in Neural Information Processing Systems, 33, 20802-20812.

**Strengths And Weaknesses:**

The manuscript is, in my opinion, well-written and accessible to the audience of TMLR. The introduction is simple to read, and a thorough background is given on the notion of statistical depth, and the traditional properties of depth functions. The theoretical results and proofs are written with sufficient care, and the numerical results are presented with sufficient computational details. Furthermore, I have read several of the theoretical results, and I am not aware of any mistakes. To the best of my reading, I therefore believe the manuscript meets the presentation and correctness standards of TMLR, and warrants publication. My remaining comments below deal with the possible impact of the proposed divergences, which I kindly ask the authors to consider in their revision.

The idea of constructing a metric between probability distributions using the Hausdorff distance between depth regions is new, to the best of my knowledge. Nevertheless, I do not think the authors have adequately substantiated why this metric is more favorable than other popular metrics, in particular the Wasserstein distance. A long line of recent optimal transport literature (cf. [1]) has argued that optimal transport maps can be used to define quantile functions in general dimension, thus there is a heuristic sense in which the Wasserstein distance is connected to the comparison of multivariate quantiles, similarly to the divergence of the authors. Furthermore, there exist trimmed analogues of the Wasserstein distance [2-3], which have a trimming constant that plays a similar role as the parameter \epsilon appearing in the current manuscript. These metrics share similar robustness properties as the divergence of the authors. There also exists a trimmed analogue of the Sliced Wasserstein distance [4], which is not only robust but can also be computed easily. It is reasonable to expect that these trimmed optimal transport metrics share similar qualitative benefits as the proposed divergence, thus I encourage the authors to expand their discussion on its possible advantages.

[1] Hallin, M. (2022). Measure transportation and statistical decision theory. Annual Review of Statistics and Its Application, 9, 401-424.

[2] Munk, A., & Czado, C. (1998). Nonparametric validation of similar distributions and assessment of goodness of fit. Journal of the Royal Statistical Society Series B: Statistical Methodology, 60(1), 223-241.

[3] Alvarez-Esteban, P. C., Del Barrio, E., Cuesta-Albertos, J. A., & Matran, C. (2008). Trimmed comparison of distributions. Journal of the American Statistical Association, 103(482), 697-704.

[4] Manole, T., Balakrishnan, S., & Wasserman, L. (2022). Minimax confidence intervals for the sliced wasserstein distance. Electronic Journal of Statistics, 16(1), 2252-2345.

---

> ### Author Response · Authors · 2024-01-21
> **Answer to Reviewer vcJR**
>
> We thank the reviewer for their constructive comments, that help improving the paper, and for acknowledging our efforts to validate the approach. We believe that our current work is helpful to the community. Regarding the concerns of the reviewer, we answered point by point below. We also added new experiments to strengthen our claims and corrected suggested typos in the new version.
>
>
> “As per my previous comment, I encourage the authors to expand their discussion of the possible benefits of their metric as compared to (trimmed) Wasserstein-type metrics. "
>
> * **Regarding flexibility.**
> One of the main benefits of our pseudo-metric is its flexibility. Our general definition allows the use of any depth function, see e.g. [6] for a review of the main depths,  at the price of choosing one that is relevant for the underlying data.
>
> * **Robustness.**
> The trimming improves the robustness of $DR_{p,\varepsilon}$ and sliced-Wasserstein. While it is not the only source of robustness in $DR_{p,\varepsilon}$, the Wasserstein metric is generally known to be non-robust [7, 8].
> Indeed, the robustness in $DR_{p,\varepsilon}$ also comes from the robustness of the chosen depth and the trimming step may not be helpful in specific situations.
>
> * **Regarding the trimming aspect.**
>   The trimming occurs in different spaces for the trimmed sliced Wasserstein and $DR_{p, \varepsilon}$. Our pseudo-metric trimming relies on peeling the larger quantile regions  (depth regions) that directly consider the multivariate data's structure. In contrast, the trimming in SW occurs at the projection level. Since, in practice, the projections are chosen uniformly on the unit sphere, this trimming does not consider the correlation of the data, for example. It may remove non-outlier points of the data. See the following paragraph and Section D.8. of the Appendix for more details.
> This discussion has been added to the Appendix, Section A.4.
>
>
> ‘‘I also encourage the authors to consider using the trimmed Wasserstein metric (or its sliced, max-sliced, etc. versions) in their numerical comparisons. I believe this would provide a more fair comparison in Figure 5. ''
>
>
> * We agree and thank the reviewer for pointing out the trimmed version of the Sliced-Wasserstein (TSW). We added the trimmed sliced Wasserstein distance in the robustness comparison of Figure 5 (with the same trimming parameter as $DR_{p, \varepsilon}$). We also tried to add this trimmed version to the experiment depicted in Figures 3 and 4. However, the values of this trimmed sliced Wasserstein appeared to not converge for our value of $\varepsilon$ and couldn’t be displayed on these plots. To investigate this surprising phenomenon, we proposed an experiment in Section D.8 of the Appendix investigating the trimming behavior of $DR_{p,\varepsilon})}$ and $TSW_{p, \varepsilon}$. The trimming in the sliced Wasserstein metric appears to drastically change the values of the metric, even for data where trimming shouldn’t have such an effect (non-standard Gaussian distribution). The trimming effect in TSW, which appears at a univariate projection level, is suboptimal. Indeed, since the integral on the unit sphere is approximated by Monte-Carlo in practice, this leads to trimming data only in some randomly chosen direction, not considering the distribution structure. In contrast, the trimming effect in $DR_{p, \varepsilon}$ is at the multivariate level on the quantile regions defined by the chosen depth functions, and therefore, trim data w.r.t. the geometry/structure of the multivariate distribution.

---

> ### Author Response · Authors · 2024-01-21
> **Follow up Answer Reviewer vcJR**
>
> “ On page 8, the authors comment that "deriving theoretical finite-sample analysis may appear to be challenging" --- could the authors elaborate on this comment? What makes it challenging to analyze estimators of the DR metric? "
>
> * The difficulty mainly relies on two aspects. The first one is controlling the deviations of the empirical depth regions w.r.t. the population versions, i.e. $d_{\mathcal{H}}(\widehat{D}^{\alpha(\beta)}, D^{\alpha(\beta)})$. It is generally difficult, notably by passing conditions on distributions to conditions on depth distributions, and has not been tackled in the data-depth community. The only rate available is the one of the halfspace depth  [9], with particular assumptions, and it is already very technical, involving arguments from stochastic geometry. In addition, for $DR_{p, \varepsilon}$, the quantile value $\alpha(\beta))$ is not deterministic. Therefore, we rather control the deviations between $d_{\mathcal{H}}(\widehat{D}^{\hat{\alpha}(\beta)}, D^{\hat{\alpha}(\beta)})$, that leads to even more technical difficulties.
>
> “On page 9, could the authors please elaborate on their comment that the sliced-Wasserstein Monte Carlo approximation degrades exponentially in the dimension? Reference [5, Theorem 6] would suggest otherwise."
>
> * **Regarding the exponential rate of the Sliced-Wasserstein Monte Carlo approximation**
> First, we agree that this claim is, at this moment, hypothetical and has been removed from the new version of the paper. To our knowledge, this question is still an open challenge and we would like to extend the discussion with the reviewer.
>
>
> *  **From the theoretical side.**
> To our knowledge, it hasn’t been proved that the Monte Carlo approximation of the Sliced-Wasserstein doesn’t depend exponentially on the dimension. We thank the reviewer for pointing out the reference [5]. However, the right side of the bound in Theorem 6 in [5] still has an integral over the unit sphere $\mathbb{S}^{d-1}$. It is difficult to see why this bound would suggest otherwise in this state. May the reviewer elaborate with this?
>
> * **From the empirical side.**
> The experiment in Figure 2 (a) of [5] would support empirically that this approximation does not depend on the dimension. The authors investigated numerically the Monte-Carlo approximation w.r.t. the number of projections on standard Gaussian distribution. The approximation error is computed with a ground truth chosen as $\widehat{SW}^{\text{MC}}$ with $10000$ projections. However, their empirical results would remain even if the Monte-Carlo approximation degrades exponentially with the dimension. Indeed, $| \hat{SW}\_{1000}^{\text{MC}} - \hat{SW}\_{10000}^{\text{MC}} | $ would still be very small even if $| \widehat{SW}_K^{\text{MC}} - \widehat{SW} |$ would converge very slowly w.r.t. the dimension.
> We investigate this question in the following experiment, described in Section D.4. in the Appendix of the new version. We first place ourselves in the same setting as Figure 2 (a) of [5], and reproduce their results in Figure 9 (left). Second, we add the same experiment with anisotropic Gaussian, Figure 9 (middle), to avoid artifacts arising from the isotropic nature of the data. In contrast to Figure 2 (a) of [5], we plot the absolute error instead of the square. These two plots would also support the claim that the Monte-Carlo approximation is not exponential in the dimension. However, this is artificially created by comparing the MC approximation to the MC instead of the truth value.
> In the third experiment, we evaluated the Monte-Carlo approximation error w.r.t. the truth value of the max-sliced Wasserstein. In the case of two standard Gaussian distributions with mean $\mathbf{m}_1$ and $\mathbf{m}_2$ with the same covariance matrix, say $I_d$, the max-sliced Wasserstein has a closed-form equal to $||\mathbf{m}_1-\mathbf{m}_2||$. We then compute the absolute value of $\widehat{SW}^{\text{MC}} - ||m_1-m_2||$ as approximation error. Of course, the statistical error remains since
> $| \widehat{SW}^{\text{MC}} - ||m_1-m_2|| | \leq || \widehat{SW}^{\text{MC}}- \widehat{SW} | + | \widehat{SW} - ||m_1-m_2|| |$. Therefore, we increased the number of samples to 10000 to minimize,  as much as possible, the statistical error. We repeated this experiment 10 times and reported $\widehat{SW}^{\text{MC}} - ||m_1-m_2||$ on Figure 9 (right). From this experiment we can clearly see an exponential behavior w.r.t. the dimension, that may come from the approximation error.
>
> **References:**
>
> [6] Mosler, K., & Mozharovskyi, P. (2022). Choosing among notions of multivariate depth statistics. Statistical Science, 37(3), 348-368.
>
>
> [7] Balaji, Y., Chellappa, R., & Feizi, S. (2020). Robust optimal transport with applications in generative modeling and domain adaptation. Advances in Neural Information Processing Systems, 33, 12934-12944.

---

> ### Author Response · Authors · 2024-01-21
> **End of the answer to Reviewer vcJR**
>
> [8] Mukherjee, D., Guha, A., Solomon, J. M., Sun, Y., & Yurochkin, M. (2021, July). Outlier-robust optimal transport. In International Conference on Machine Learning (pp. 7850-7860). PMLR.
>
>
> [9] Brunel, V. E. (2019). Concentration of the empirical level sets of Tukey’s halfspace depth. Probability Theory and Related Fields, 173(3-4), 1165-1196.

---

> > ### Comment · Reviewer_vcJR · 2024-01-22
> > **Response to authors' revision**
> >
> > I would like to thank the authors for thoroughly addressing my comments. I find it remarkable to see the difference in the imapct of the trimming parameter between TSW and DR. I agree with the authors that this can be explained by the fact that the DR trimming occurs in a difference space than for TSW, and I believe this points to an interesting qualitative difference between these metrics. I appreciate their pointing this out.
> >
> > Regarding the Monte-Carlo discussion, I think there is a confusion between integral-slicing and max-slicing. My comment was about integral-slicing, which is also the setting of the reference [5] that I gave. In this context, I disagree that the dependence on dimension should be exponential, at least for compactly-supported measures. Indeed, let $\mu$ and $\nu$ be compactly-supported measures, and suppose we wish to estimate $\text{SW}_2(\mu,\nu)$ via Monte-Carlo (ignoring statistical error). Then, as per [5, Theorem 6], the dimension-dependence is governed by
> >
> > $$V_d = Var_\theta[W_2(\mu_\theta, \nu_\theta)]$$
> >
> > where the expectation is over a uniformly-distributed random variable $\theta$ over $\mathbb{S}^{d-1}$, and where $\mu_\theta$ and $\nu_\theta$ are the projections of $\mu$ and $\nu$ along the direction $\theta$. Notice that the above is bounded above by
> >
> > $$V_d \leq E_\theta E_{X,Y} \|X^\top\theta - Y^\top\theta\|^2,$$
> >
> > where we can take the second expectation over independent random variables $X \sim \mu$ and $Y \sim \nu$. Using the compact support of the measures, one deduces
> >
> > $$V_d \lesssim E_\theta ||\theta||^2 \lesssim d.$$
> >
> > On the other hand, for the max-sliced distance, I fully agree that it is plausible for the dependence on dimension to be exponential.
> >
> > Barring any possible changes that the authors may wish to make to the text based on this discussion, I believe the paper is in good shape, and I recommend acceptance.

---

> ### Author Response · Authors · 2024-01-23
> **Response to Reviewer vcJR**
>
> We would like to thank you for your reactivity as well as the careful review of our answer and the new version of the paper.
>
> Regarding the Monte Carlo approximation discussion, we thank the reviewer for elaborating on the implications of Theorem 6 of [5]. We agree with the reviewer. We were confusing sliced Wasserstein metrics with maximum sliced Wasserstein metrics. This is now clear.
>
>
> Therefore, we removed the claim of exponential dependency of the sliced Wasserstein on the new version. We also removed the additional experiment on max-sliced Wasserstein, which has poor relevance for the paper and could likely induce confusion between rates of sliced and max-sliced Wasserstein metrics.

---

### Decision · Action_Editor_wECb · 2024-02-06

**Recommendation:** Accept as is

**Comment:**

All the reviewers and myself found the paper interesting and should be accepted.

**Audience:**

The design of new and robust pseudo-metrics on the space of probability distributions is a topic of interest for the community since it is at the core of many existing methods in machine learning. Therefore, the potential audience for this research is extensive.

**Claims And Evidence:**

In this paper, the authors introduce a new pseudo-metric on the space of probability measures. This discrepancy is defined based on the upper-level sets of convex depth functions, such as halfspace and projection depth.

The authors demonstrate that this pseudo-metric enjoys favorable properties, including invariance/translation stability, and robustness with respect to the finite sample breakdown point. Additionally, the paper presents an algorithm for the approximate computation of this pseudo-metric.

Finally, numerical experiments are conducted to analyze the effectiveness of the approximation scheme and the statistical rates associated with this pseudo-metric. The authors also empirically demonstrate the robustness of their pseudo-metric to outliers and highlight its benefits in two machine learning applications.